# Mutations in SKI in Shprintzen–Goldberg syndrome lead to attenuated TGF-β responses through SKI stabilization

Ilaria Gori[1], Roger George[2], Andrew G Purkiss[2], Stephanie Strohbuecker[3], Rebecca A Randall[1], Roksana Ogrodowicz[2], Virginie Carmignac[4], Laurence Faivre[4], Dhira Joshi[5], Svend Kjær[2], Caroline S Hill[1]*

[1]Developmental Signalling Laboratory, The Francis Crick Institute, London, United Kingdom; [2]Structural Biology Facility, The Francis Crick Institute, London, United Kingdom; [3]Bioinformatics and Biostatistics Facility, The Francis Crick Institute, London, United Kingdom; [4]INSERM - Université de Bourgogne UMR1231 GAD, FHU-TRANSLAD, Dijon, France; [5]Peptide Chemistry Facility, The Francis Crick Institute, London, United Kingdom

**Abstract** Shprintzen–Goldberg syndrome (SGS) is a multisystemic connective tissue disorder, with considerable clinical overlap with Marfan and Loeys–Dietz syndromes. These syndromes have commonly been associated with enhanced TGF-β signaling. In SGS patients, heterozygous point mutations have been mapped to the transcriptional co-repressor SKI, which is a negative regulator of TGF-β signaling that is rapidly degraded upon ligand stimulation. The molecular consequences of these mutations, however, are not understood. Here we use a combination of structural biology, genome editing, and biochemistry to show that SGS mutations in SKI abolish its binding to phosphorylated SMAD2 and SMAD3. This results in stabilization of SKI and consequently attenuation of TGF-β responses, both in knockin cells expressing an SGS mutation and in fibroblasts from SGS patients. Thus, we reveal that SGS is associated with an attenuation of TGF-β-induced transcriptional responses, and not enhancement, which has important implications for other Marfan-related syndromes.

*For correspondence:
caroline.hill@crick.ac.uk

**Competing interests:** The authors declare that no competing interests exist.

## Introduction

Shprintzen–Goldberg syndrome (SGS) is a multisystemic connective tissue disorder. Common features observed in SGS patients include craniofacial, skeletal, and cardiovascular anomalies, ranging from heart valve defects to thoracic aortic aneurysms, all of which are also characteristic of Marfan syndrome (MFS) and Loeys–Dietz syndrome (LDS) (*Cook et al., 2015a*; *Verstraeten et al., 2016*; *Loeys et al., 2005*; *Williams et al., 2007*). In addition, SGS patients present with craniosynostosis, intellectual disability, and skeletal muscle hypotonia (*Shprintzen and Goldberg, 1982*; *Glesby and Pyeritz, 1989*; *Greally et al., 1998*). All three syndromes have been linked to deregulation of the transforming growth factor β (TGF-β) signaling pathway (*Cannaerts et al., 2015*).

The TGF-β family of ligands comprises the TGF-βs themselves, Activins, Nodal, bone morphogenetic proteins (BMPs), and growth differentiation factors (GDFs) and they play pleiotropic roles in embryonic development and tissue homeostasis. In addition, their signaling is deregulated in diverse pathologies (*Miller and Hill, 2016*). They exert their action by binding to type I and type II serine/threonine kinase receptors at the cell surface (TGFBR1 and TGFBR2, respectively, for the TGF-βs) (*Massagué, 2012*). In the resulting ligand-bound heterotetrameric receptor complex, the type II receptor phosphorylates and activates the type I receptor, which in turn phosphorylates the intracellular mediators, the receptor-regulated SMADs (R-SMADs). Once phosphorylated, the R-SMADs

(SMAD2 and SMAD3 in the case of TGF-β, Activin, and Nodal) associate with the common mediator of the pathway, SMAD4. The resulting heterotrimeric complexes accumulate in the nucleus where they interact with other transcriptional regulators to activate or repress target gene expression (*Massagué, 2012*). Two highly related co-repressors, SKI and SKIL (formerly known as SnoN), act as negative regulators in the pathway (*Deheuninck and Luo, 2009*; see below).

The role of deregulated TGF-β signaling in Marfan-related syndromes is controversial. MFS is caused by loss-of-function mutations in the extracellular matrix protein, Fibrillin 1 (FBN1) (*Dietz et al., 1991*). These mutations are thought to increase the bioavailability of TGF-β ligands, as FBN1 binds the latent form of the TGF-βs (*Neptune et al., 2003*; *Kaartinen and Warburton, 2003*). Supporting the idea that excessive TGF-β signaling contributes to the manifestations of MFS, a TGF-β neutralizing antibody significantly improved the lung phenotype in a mouse model of MFS (homozygous Fbn1$^{mgΔ}$) (*Neptune et al., 2003*; *Cannaerts et al., 2015*) and reduced the occurrence of aortic aneurysms in the Fbn1$^{C1039G/+}$ mouse model of MFS (*Habashi et al., 2006*). Contradicting these results, others have shown that the aortopathy in the Fbn1$^{C1039G/+}$ mouse model is not mediated by excessive TGF-β signaling and in fact is exacerbated by loss of TGF-β signaling in smooth muscle cells (*Wei et al., 2017*). Furthermore, TGF-β signaling protects against abdominal aortic aneurysms in angiotensin II-infused mice (*Angelov et al., 2017*). This controversy emphasizes the importance of understanding exactly how TGF-β signaling is impacted in MFS. Furthermore, the related syndrome LDS is caused by pathogenic mutations in several different components of the TGF-β pathway, TGFBR1, TGFBR2, SMAD2, SMAD3, and the ligands, TGFB2 and TGFB3. These mutations all cause missense amino acid substitutions that have been either verified in vitro, or are predicted to be loss of function, implying that LDS is caused by attenuated TGF-β signaling (*Horbelt et al., 2010*; *Cardoso et al., 2012*; *Schepers et al., 2018*). However, paradoxically, histological and biochemical studies of aortic tissue derived from LDS patients reveal an apparent high TGF-β signaling signature (*van de Laar et al., 2012*; *Gallo et al., 2014*; *Lindsay et al., 2012*). SGS is caused by mutations in SKI, and both SMAD-mediated and non-SMAD-mediated TGF-β signaling has been reported to be increased in primary dermal fibroblasts from SGS patients (*Doyle et al., 2012*).

The co-repressors SKI and SKIL play important roles in a number of different cellular processes including proliferation, differentiation, transformation, and tumor progression (*Bonnon and Atanasoski, 2012*). They are dimeric proteins that interact with both phosphorylated SMAD2 and SMAD3 (PSMAD2 or PSMAD3) via short motifs at their N-termini, and with SMAD4 via a SAND domain (named after Sp100, AIRE-1, NucP41/75, DEAF-1) in the middle of both proteins (*Deheuninck and Luo, 2009*). Between these two domains lies a Dachshund homology domain (DHD), which is thought to also be important for R-SMAD binding (*Wilson et al., 2004*; *Ueki and Hayman, 2003*). SKI and SKIL both contain a leucine zipper domain in their C-termini, through which they dimerize (*Deheuninck and Luo, 2009*). They are negative regulators of TGF-β/Activin signaling, with two distinct mechanisms of regulation having been proposed. In one model, SKI and SKIL bind with SMAD4 to SMAD binding elements (SBEs) of TGF-β/Activin target genes, and recruit co-repressors such as NCOR1 or SIN3A (*Tokitou et al., 1999*; *Nomura et al., 1999*; *Stroschein et al., 1999*; *Deheuninck and Luo, 2009*). They thus maintain the transcription of these target genes suppressed in the absence of signal. Upon TGF-β/Activin signaling, SKI and SKIL are rapidly degraded by the E3 ubiquitin ligase, RNF111 (formerly known as Arkadia), a process that requires SKI/SKIL binding to PSMAD2 or PSMAD3 (*Le Scolan et al., 2008*; *Levy et al., 2007*; *Nagano et al., 2007*). This then allows the activated SMAD3–SMAD4 complexes to bind the exposed SBEs and activate target gene transcription (*Levy et al., 2007*; *Stroschein et al., 1999*). In the competing model, SKI and SKIL act as repressors of active signaling simply by binding to PSMAD2 or PSMAD3 and SMAD4 in such a way as to disrupt the activated PSMAD2/PSMAD3–SMAD4 complexes (*Luo, 2004*; *Ueki and Hayman, 2003*; *Wu et al., 2002*). The heterozygous missense mutations that cause SGS have been mapped in SKI to the N-terminal R-SMAD-binding domain, with some small deletions and point mutations also found in the DHD, which is also necessary for R-SMAD binding (*Carmignac et al., 2012*; *Doyle et al., 2012*; *Schepers et al., 2015*). Thus, depending on the mechanism whereby SKI inhibits TGF-β/Activin signaling, loss of the interaction with PSMAD2/PSMAD3 would be predicted to have opposite effects on signaling output. If the PSMAD2/PSMAD3 interaction is required for SKI degradation, its loss would inhibit TGF-β signaling. However, if SKI binding to PSMAD2/PSMAD3 disrupts active SMAD complexes, then its loss would promote TGF-β signaling.

Here we use a combination of genome editing, structural biology, biochemistry, and analysis of patient samples to elucidate the molecular mechanism underlying SGS and to resolve the paradox surrounding the role of TGF-β signaling in Marfan-related syndromes. We first determine at the molecular level how SKI/SKIL function in the TGF-β/Activin signaling pathways and show that an intact ternary phosphorylated R-SMAD–SMAD4 complex is required for ligand-induced SKI/SKIL degradation. We demonstrate that the SGS mutations in SKI abolish interaction with PSMAD2 and PSMAD3 and this results in an inability of SKI to be degraded in response to TGF-β/Activin signaling. We go on to show that SKI stabilization results in an attenuation of the TGF-β transcriptional response in both knockin HEK293T cells and fibroblasts from SGS patients. Our work unequivocally establishes that SGS mutations lead to an attenuated TGF-β response, which has major implications for all the Marfan-related syndromes.

## Results

### A PSMAD2/3–SMAD4 ternary complex is essential for TGF-β/Activin-induced degradation of SKI/SKIL

To understand the consequences of SKI mutations in SGS and to resolve the paradox surrounding the function of TGF-β signaling in Marfan-related syndromes, we first set out to determine exactly how SKI and SKIL act as negative regulators of TGF-β and Activin signaling. We and others have previously demonstrated that SKI and SKIL are rapidly degraded upon TGF-β/Activin stimulation by the E3 ubiquitin ligase RNF111, and this requires PSMAD2 or PSMAD3 (*Le Scolan et al., 2008*; *Levy et al., 2007*; *Nagano et al., 2007*). Knockdown experiments suggested that SMAD4 was not necessary (*Levy et al., 2007*), but we subsequently showed that tumor cells deleted for SMAD4 or containing mutations in SMAD4 that abolish interactions with activated R-SMADs, abrogated TGF-β-induced degradation of SKI/SKIL (*Briones-Orta et al., 2013*). Whether the requirement for SMAD4 was direct or indirect was not clear.

To define the role of SMAD4 in SKI/SKIL degradation, we used CRISPR/Cas9 technology to delete SMAD4 in transformed embryonic kidney cells HEK293T, which express both SKI and SKIL, and in the human keratinocyte cell line, HaCaT, which predominantly express SKIL (*Levy et al., 2007*; *Figure 1—source data 1*). In wild-type (WT) cells, TGF-β/Activin induced rapid SKI and SKIL degradation, compared to cells treated with the TGFBR1 inhibitor, SB-431542 (*Inman et al., 2002*; *Figure 1A,B*). Deletion of SMAD4 in multiple clones of both cell types abolished ligand-induced SKI/SKIL degradation (*Figure 1A,B*). We validated these SMAD4-null cell lines by demonstrating that transient expression of SMAD4 could rescue TGF-β/Activin-induction of the SMAD3–SMAD4 reporter, CAGA$_{12}$-Luciferase (*Figure 1—figure supplement 1A,B*). Furthermore, we could show that loss of SMAD4 inhibited the ligand-induced expression of a number of endogenous TGF-β and BMP target genes (*Figure 1—figure supplement 1C*). By knocking out SMAD2 or SMAD3 individually or together, we also confirmed that these R-SMADs are absolutely required for TGF-β/Activin-induced degradation of SKI and SKIL and act redundantly (*Figure 1C*; *Figure 1—source data 1*). Thus, R-SMADs and SMAD4 are all essential for TGF-β/Activin-dependent SKI/SKIL degradation.

In addition to forming a ternary complex with PSMAD2 or PSMAD3, SMAD4 has also been shown to interact directly with SKI and SKIL through their SAND domains (*Walldén et al., 2017*; *Wu et al., 2002*). To determine which of these SMAD4 interactions were important for TGF-β/Activin-induced SKI/SKIL degradation, we stably reintroduced enhanced GFP (EGFP) fusions of WT or mutated SMAD4 into HaCaT SMAD4-null cells. We selected two missense mutations on opposite faces of the C-terminal Mad homology 2 (MH2) domain of SMAD4: Asp351->His (D351H) and Asp537–>Tyr (D537Y) (*Shi et al., 1997*). These have been shown to occur naturally in the human colorectal cancer cell lines CACO-2 and SW948, and have lost the ability to bind phosphorylated R-SMADs (*De Bosscher et al., 2004*). In addition, we used the crystal structure of the MH2 domain of SMAD4 and the SAND domain of SKI, to design two mutations Ala433–>Glu (A433E) and Ile435–>Tyr (I435Y), that would be expected to abolish SMAD4 binding to SKI and SKIL (*Wu et al., 2002*).

We confirmed that these SMAD4 mutants behaved as expected in the rescue cell lines by testing their interaction with SKIL and R-SMADs by immunoprecipitation. As endogenous RNF111 triggers SKIL degradation in TGF-β/Activin-dependent manner, the stable SMAD4-expressing HaCaT rescue cell lines were incubated with the proteasome inhibitor, MG-132 for 3 hr prior to TGF-β stimulation,

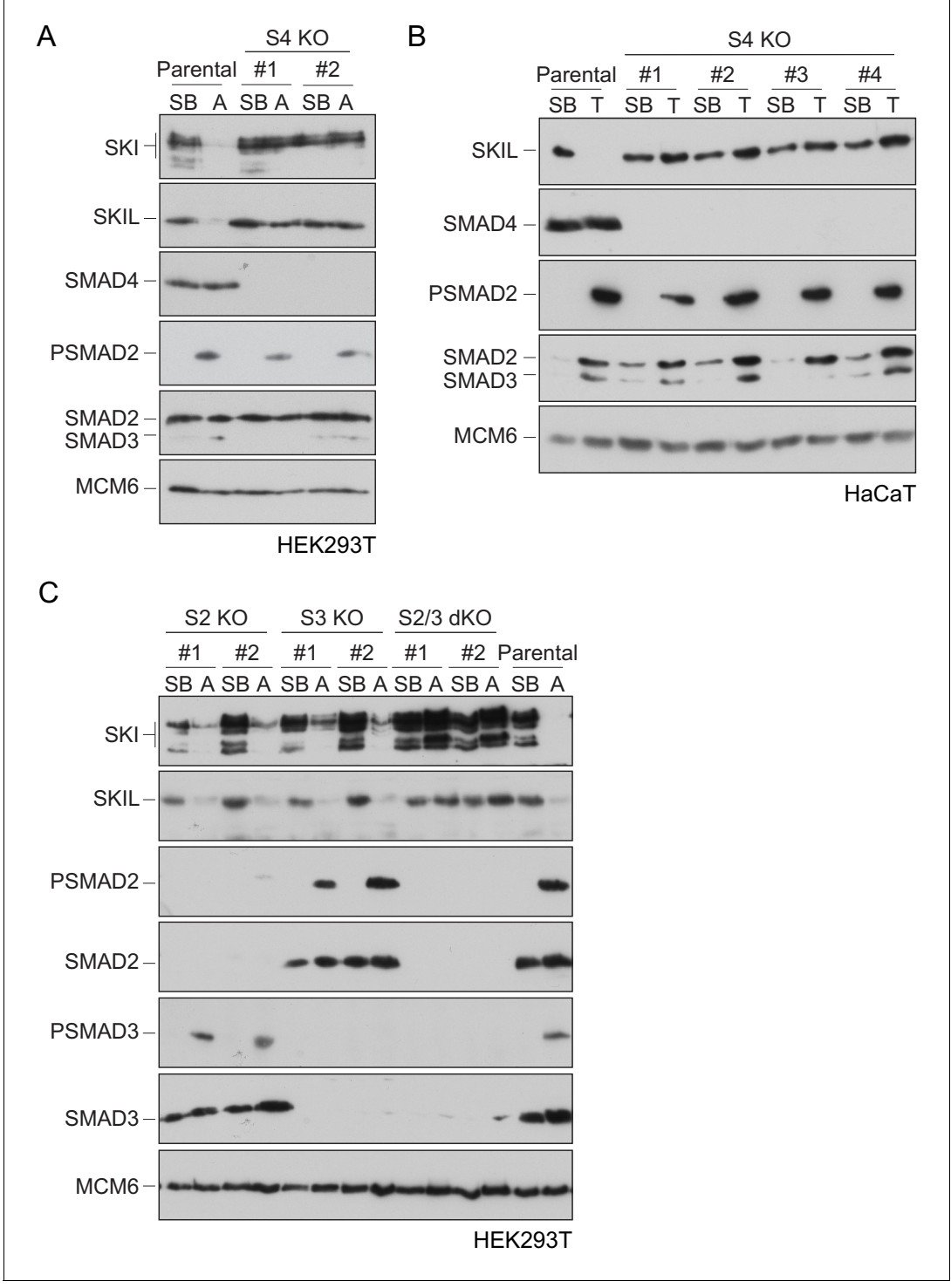

**Figure 1.** Requirement of SMAD2 or SMAD3 and SMAD4 for SKI and SKIL degradation. (**A and C**) The parental HEK293T cell line and two individual SMAD4 knockout clones (**A**) or two individual SMAD2, SMAD3 knockout clones, or two SMAD2 and SMAD3 double knockout clones (**C**) were incubated overnight with 10 µM SB-431542, washed out, then incubated with full media containing either SB-431542 or 20 ng/ml Activin A for 1 hr, as indicated. Whole-cell extracts were immunoblotted with the antibodies indicated. (**B**) Parental HaCaT and four individual SMAD4 knockout clones were treated as above, except that they were treated with 2 ng/ml TGF-β for 1 hr instead of Activin A. Nuclear lysates were immunoblotted using the antibodies indicated. SB, SB-431542; A, Activin A; T, TGF-β; S2, SMAD2; S3, SMAD3; S2/3, SMAD2 and SMAD3; S4, SMAD4; KO, knockout; dKO, double knockout. The online version of this article includes the following source data and figure supplement(s) for figure 1:

**Source data 1.** Sequences of knockout alleles made in HEK293T cells.

*Figure 1 continued on next page*

*Figure 1 continued*

**Figure supplement 1.** SMAD4 is essential for TGF-β/Activin-induced transcriptional responses.
**Figure supplement 1—source data 1.** Luciferase assay data for HEK293T S4 KO clones, as presented in *Figure 1—figure supplement 1A*.
**Figure supplement 1—source data 2.** Luciferase assay data for HaCaT S4 KO clones, as presented in *Figure 1—figure supplement 1B*.
**Figure supplement 1—source data 3.** qPCR data for HaCaT S4 KO clones, as presented in *Figure 1—figure supplement 1C*.

to block SKIL degradation. As predicted, the D351H and D537Y SMAD4 mutants had lost their ability to bind SMAD2 upon TGF-β induction, but retained the interaction with SKIL. By contrast, A433E and I435Y SMAD4 mutants were unable to bind SKIL, but could interact with SMAD2 upon TGF-β stimulation (*Figure 2A*). Furthermore, as expected, D351H and D537Y SMAD4 mutants failed to rescue the ability of TGF-β to induce expression of CAGA$_{12}$-Luciferase in HaCaT SMAD4-null cells or rescue TGF-β-induced transcription of target genes, but the A433E and I435Y SMAD4 mutants rescued these responses almost as well as WT SMAD4 (*Figure 2—figure supplement 1A,B*).

Having demonstrated that these mutants behaved as designed, we asked which were able to mediate TGF-β-induced SKIL degradation, using three different assays. In a Western blot assay using nuclear extract, we found that reintroduction of WT SMAD4 in SMAD4-null cells caused a 50% reduction in SKIL levels in TGF-β-induced cells compared to those treated with SB-431542 (*Figure 2B*). However, none of the four SMAD4 mutants could rescue TGF-β-induced SKIL degradation (*Figure 2B*). We then established a flow cytometry assay to quantify SKIL protein stability in EGFP/EGFP-SMAD4-expressing cells (*Figure 2C*; *Figure 2—figure supplement 1C*). Treatment with TGF-β for 1 hr caused a 52% reduction in the relative median fluorescence intensity in the EGFP-SMAD4 WT-expressing cells, reflecting SKIL levels, compared to cells treated with SB-431542 (*Figure 2C*). However, for all four SMAD4 mutants tested, the median fluorescence was not decreased by TGF-β treatment (*Figure 2C*). Finally, we used an immunofluorescence analysis to monitor SKIL protein stability following TGF-β exposure. SMAD4-null cells showed strong nuclear staining of SKIL in the non-signaling condition (SB-431542), which remained unchanged by TGF-β treatment (*Figure 3*). Reintroduction of WT EGFP-SMAD4 conferred the ability to degrade SKIL upon TGF-β treatment, whereas none of the mutant SMAD4s were able to rescue SKIL degradation (*Figure 3*, arrows). Thus, all three assays demonstrate that a ternary R-SMAD–SMAD4 complex is absolutely necessary for TGF-β-induced SKIL degradation, as is the ability of SMAD4 to interact with SKIL itself. This suggests that within a canonical activated ternary SMAD complex, the R-SMAD component binds to the N-terminal region of SKIL/SKI, whilst SMAD4 binds the SAND domain, and both interactions are absolutely required for SKIL/SKI degradation.

## SGS mutations inhibit the interaction of SKI with phosphorylated R-SMADs

We next investigated the consequences of the SGS mutations on SKI and SKIL's ability to interact with the R-SMADs. SKI and SKIL share a highly conserved region at their N-terminus comprising the domain known to be important for R-SMAD binding (*Deheuninck and Luo, 2009*; *Figure 4—figure supplement 1A*). We first determined the minimal region of SKI required for R-SMAD binding using peptide pulldown assays with biotinylated SKI peptides and whole-cell extract from uninduced and TGF-β-treated HaCaT cells. This revealed that amino acids 11–45 of SKI are sufficient for binding to PSMAD2 and PSMAD3 upon TGF-β stimulation, whilst the unphosphorylated SMADs did not bind to any of the SKI peptides (*Figure 4—figure supplement 1B*). SMAD4 is also pulled down in these assays in a ligand-induced manner, by virtue of its interaction with the phosphorylated R-SMADs.

The SGS mutations discovered so far mostly cluster within this 11–45 region of SKI and a few deletions and point mutations have additionally been mapped in the DHD domain (*Carmignac et al., 2012*; *Doyle et al., 2012*; *Schepers et al., 2015*). The residues mutated are completely conserved, both between species, and also in the related protein, SKIL (*Figure 4—figure supplement 1A*; *Carmignac et al., 2012*). To determine the effect of these mutations on R-SMAD interaction, we introduced six different SGS mutations into the SKI peptide 11–45 and showed that they all prevented binding of PSMAD2 and PSMAD3, and as a result, also SMAD4 (*Figure 4A*). These results were also confirmed with the equivalent mutations in SKIL (*Figure 4B*). We proved that the interaction with SMAD2 was mediated via its MH2 domain using a mouse embryonic fibroblast

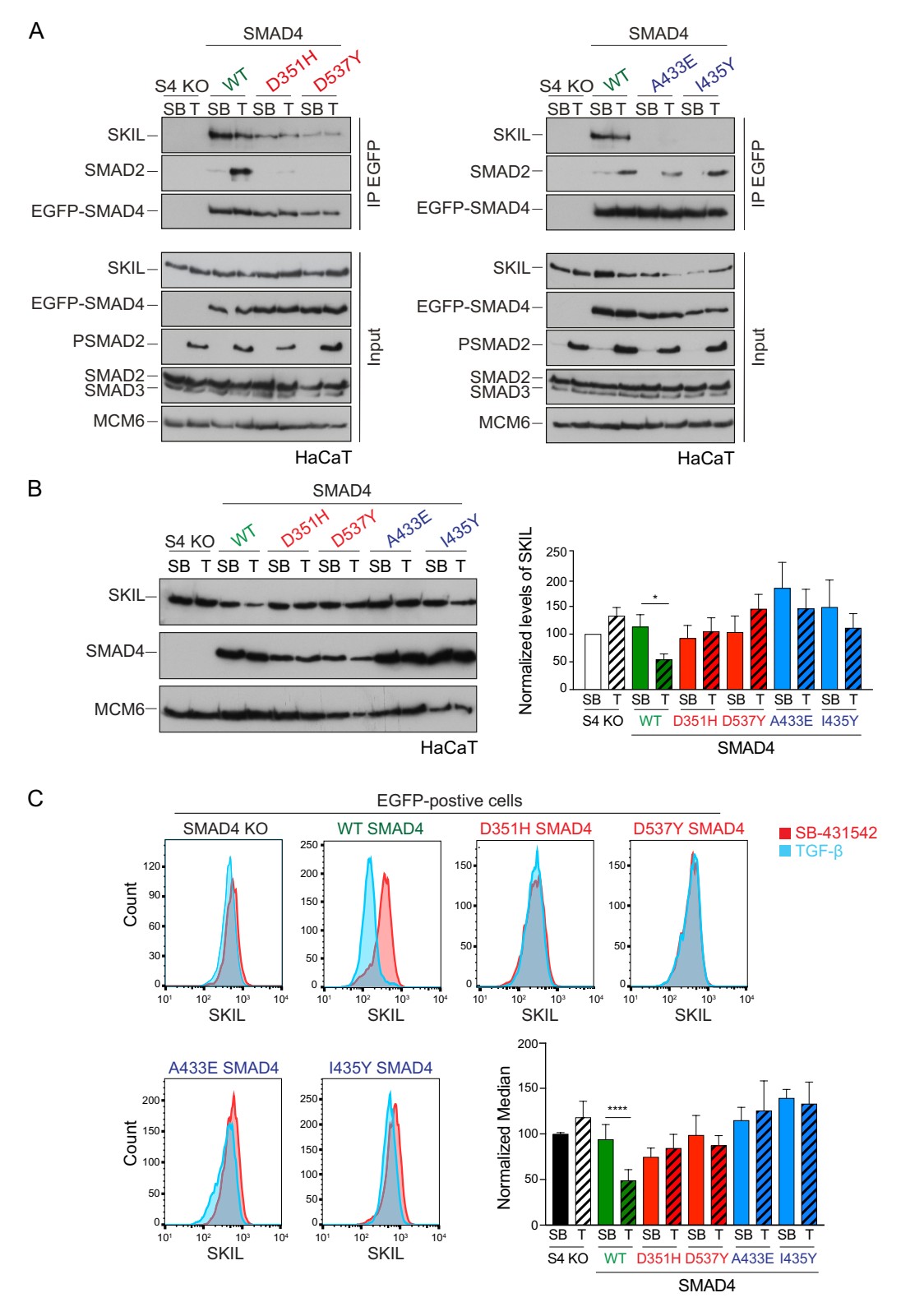

**Figure 2.** Characterization of the role of SMAD4 in TGF-β-induced SKIL degradation. (**A–C**) HaCaT SMAD4 knockout (S4 KO) cells were stably transfected with EGFP alone, or EGFP SMAD4 (WT) or with four different EGFP-SMAD4 mutants (D351H, D537Y, which abolish interaction with the R-SMADs, and A433E and I435Y, which do not interact with SKIL). (**A**) Cells were incubated overnight with 10 μM SB-431542, washed out and pre-incubated with 25 μM MG-132 for 3 hr, and then treated either with 10 μM SB-431542 or 2 ng/ml TGF-β for 1 hr. Whole-cell extracts were

*Figure 2 continued on next page*

*Figure 2 continued*

immunoprecipitated (IP) with GFP-trap agarose beads. The IPs were immunoblotted using the antibodies shown. Inputs are shown below. (**B**) Nuclear lysates were prepared from the HaCaT S4 KO cells stably transfected with EGFP alone or with EGFP-SMAD4 constructs as indicated, treated as in (**A**), but without the MG-132 step and immunoblotted using the antibodies shown. On the right the quantifications are the normalized average ± SEM of five independent experiments. The quantifications are expressed as fold changes relative to SB-431542-treated S4 KO cells. (**C**) Levels of SKIL in the EGFP-positive S4 KO rescue cell lines treated as in (**B**), assayed by flow cytometry. Each panel shows an overlay of the indicated treatment conditions. The red line indicates the SB-431542-treated sample, whereas the cyan line indicates the TGF-β-treated sample. Quantifications are shown bottom right. For each group, the percentage of the median fluorescence intensity normalized to the SB-431542-treated sample is quantified. Data are the mean ± SEM of five independent experiments. The p-values are from one-way ANOVA with Sidak's post hoc correction *p<0.05; ****p<0.0001. SB, SB-431542; T, TGF-β.

The online version of this article includes the following source data and figure supplement(s) for figure 2:

**Source data 1.** Quantification of Western blot for HaCaT S4 KO rescue cell lines, as presented in *Figure 2B*.

**Source data 2.** Flow cytometry data for HaCaT S4 KO rescue cell lines, as presented in *Figure 2C*.

**Figure supplement 1.** Transcriptional activity of the SMAD4 mutants compared to WT SMAD4.

**Figure supplement 1—source data 1.** Luciferase assay data for HaCaT S4 KO rescue cell lines, as presented in *Figure 2—figure supplement 1A*.

**Figure supplement 1—source data 2.** qPCR data for HaCaT S4 KO rescue cell lines, as presented in *Figure 2—figure supplement 1B*.

cell line that expresses a truncated SMAD2 protein comprising just the MH2 domain (*Piek et al., 2001*; *Das et al., 2009*; *Figure 4C*). We confirmed this using recombinant human phosphorylated SMAD2 MH2 domain produced in insect cells by co-expressing the SMAD2 MH2 domain with the kinase domain of TGFBR1 (*Figure 4D*). In both cases, the SGS mutations prevented interaction of the SKI peptide with the SMAD2 MH2 domain.

We next used a peptide array to gain a better understanding of which amino acids can be tolerated at the positions found to be mutated in SGS and to determine which other amino acids in this region of SKI are essential for the R-SMAD interaction. The SKI peptide corresponding to amino acids 11–45 was synthesized as an array on a cellulose sheet such that each residue in the sequence between residues 19 and 35 was substituted with all 19 alternative amino acids (*Figure 4E*; *Figure 4—source data 1*). The array was probed with a recombinant PSMAD3–SMAD4 trimer, generated by co-expressing SMAD3 and SMAD4 with the TGFBR1 kinase domain in insect cells. The PSMAD3–SMAD4 complex was then detected using a fluorescently-labeled SMAD2/3 antibody. Eight residues are intolerant to almost any amino acid substitution (Thr20, Leu21, Phe24, Ser28, Ser31, Leu32, Gly34, and Pro35). Strikingly, six of these residues are the amino acids known to be mutated in SGS patients, and the array results readily explain why these residues are mutated to a number of different amino acids in SGS (*Figure 4E*; for quantification, see *Figure 4—figure supplement 1C* and *Figure 4—source data 2*). In addition, Thr20 and Phe24 are also crucial residues for binding the PSMAD3–SMAD4 complex, but have not yet been reported as disease mutations. Mutations in the other nine amino acids do not impair the binding, and almost any other amino acid apart from proline can be tolerated at these positions.

## Crystal structure of the SKI peptide with the phosphorylated SMAD2 MH2 domain

To discover why these eight amino acids were so crucial for R-SMAD binding, and also to understand why SKI and SKIL only recognize phosphorylated R-SMADs, we solved the crystal structure of the SKI peptide (amino acids 11–45) with a phosphorylated homotrimer of the SMAD2 MH2 domain, produced in insect cells as described above. We confirmed using SEC-MALLS that the phosphorylated SMAD2 MH2 domain was indeed trimeric in solution (*Figure 5—figure supplement 1A*). Analysis of the binding affinity of the SKI peptide to the SMAD2 MH2 domain trimer indicated that the dissociation constant ($K_d$) was in the low nanomolar range (*Figure 5—figure supplement 1B*). The structure was determined by molecular replacement and refined at 2 Å resolution and readily explained why the crucial amino acids identified in the peptide were required for SMAD2 binding (*Figure 5*; *Figure 5—figure supplement 1C*).

The SKI peptide binds on the outside face of the MH2 domain at the so-called three helix bundle, comprising helices 3, 4, and 5 (*Wu et al., 2001*; *Figure 5A*). The N-terminal helix of SKI packs against helix 3 of SMAD2, and the C-terminal portion of the SKI peptide, which contains the critical Gly34 and Pro35, forms a sharp turn that is stabilized by pi-stacking coordination between Phe24 of

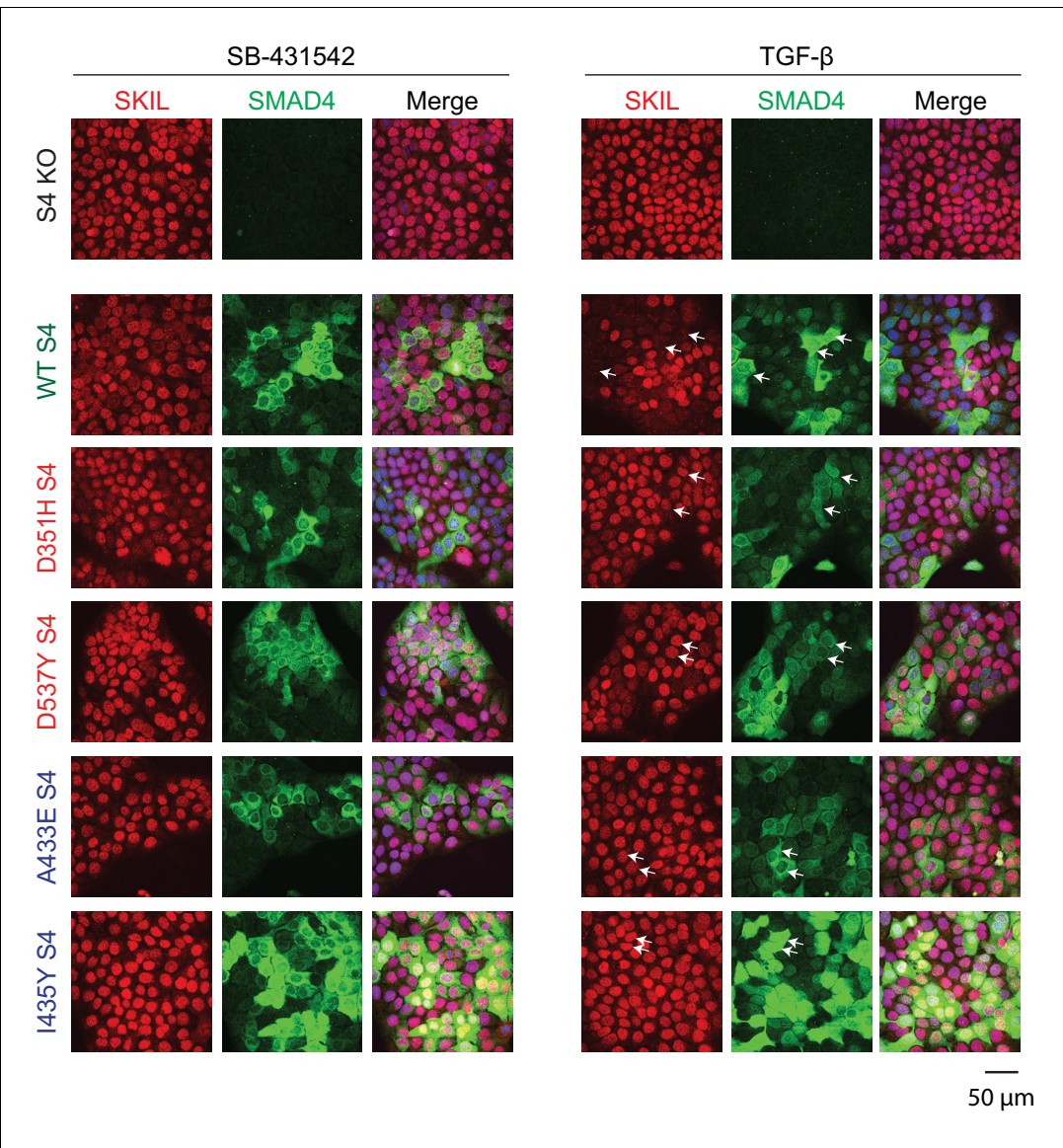

**Figure 3.** Visualization of TGF-β-induced SKIL degradation. HaCaT SMAD4 knockout (S4 KO) cells or those stably expressing EGFP SMAD4 WT or EGFP SMAD4 mutants were incubated overnight with 10 μM SB-431542, washed out, and incubated for 1 hr with 10 μM SB-431542 or with 2 ng/ml TGF-β. Cells were fixed and stained for EGFP (for SMAD4), SKIL, and with DAPI (blue) to mark nuclei and imaged by confocal microscopy. The merge combines SKIL, SMAD4, and DAPI staining. Arrows indicate examples of EGFP-expressing cells and corresponding levels of nuclear SKIL. Scale bar corresponds to 50 μm.

SKI, Trp448 of SMAD2, and Pro35 of SKI (*Figure 5B*). Moreover, the NE1 of the Trp448 side chain forms a H-bond to the main chain carbonyl group of Gly33, which in turn positions Pro35 for the interaction with Trp448 (*Figure 5B*). Furthermore, Glu270 in SMAD2 provides a pocket, which has a negatively charged base that ties down SKI Gly34 through hydrogen-bonding to its main chain amides. Other key interactions involving amino acids identified above as crucial for binding include the main chain carbonyl of Ser31, which forms a hydrogen bond to the ND1 of Asn387 in helix 3 (*Figure 5C*), and the hydroxyl group of SKI Thr20, which forms a hydrogen bond with the Gln455 at the end of helix 5 of SMAD2, and is nearly completely buried in the interface (*Figure 5D*). The two leucine residues (Leu21 and Leu32) that are mutated in SGS are both buried in the structure (*Figure 5E,F*). The structure we obtained is consistent with a SKI–SMAD2 MH2 domain structure that was published by others, while this work was in progress (*Miyazono et al., 2018*). In that case,

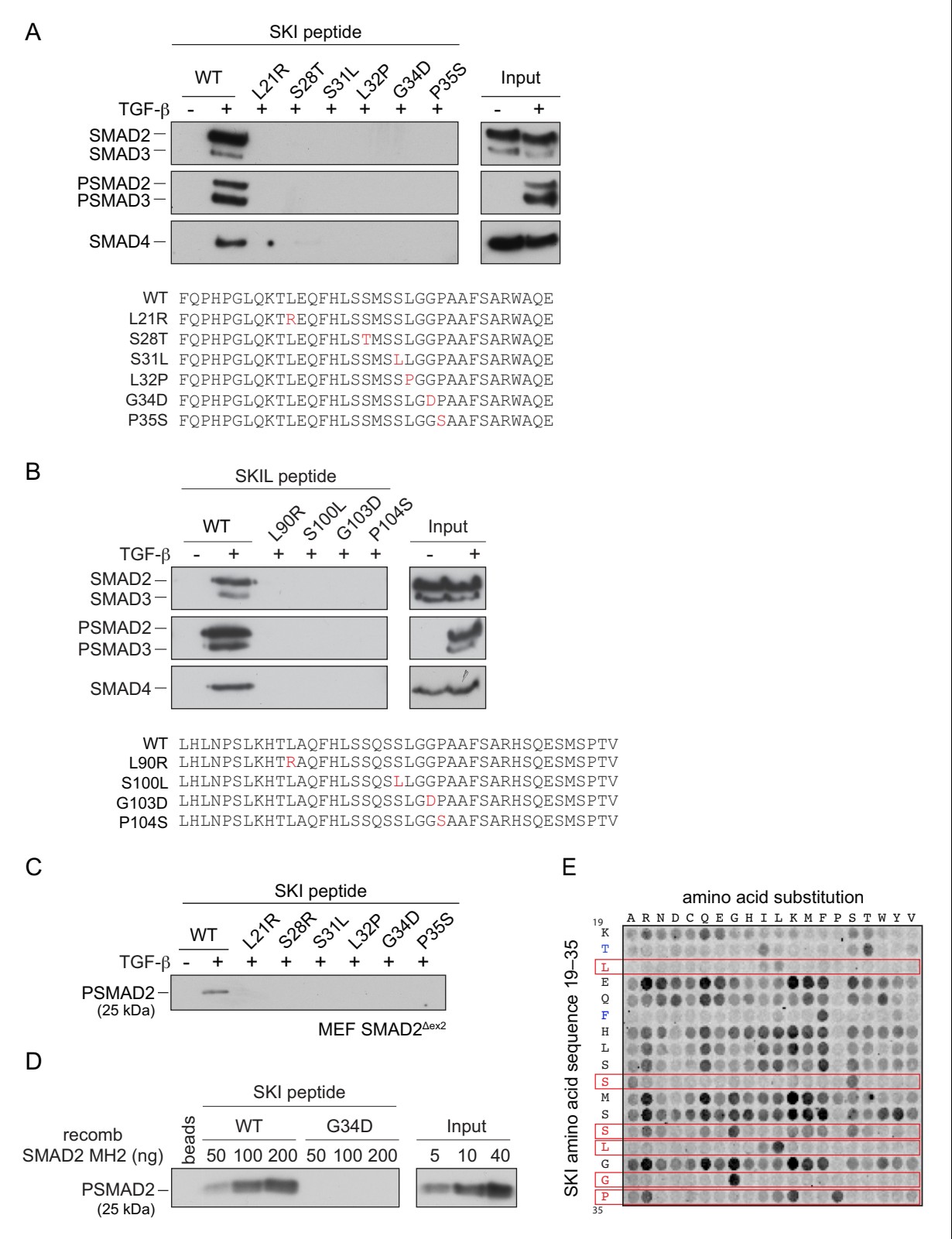

**Figure 4.** SGS mutations inhibit binding of SKI to SMAD2/3. (**A and B**) HaCaT cells were treated or not with 2 ng/ml TGF-β. A peptide pulldown assay was performed on whole cells extracts and pulldowns were immunoblotted with the antibodies indicated. Inputs are shown on the right. (**A**) Wild-type (WT) SKI peptides corresponding to amino acids 11–45 or containing SGS point mutations as shown in red were used. (**B**) WT SKIL peptides corresponding amino acids 80–120 or containing mutations (in red) corresponding to SGS mutations in SKI were used. (**C**) WT SKI peptides or those

*Figure 4 continued on next page*

*Figure 4 continued*

containing SGS point mutations were used in pulldown assays with whole-cell extracts of SMAD2-null mouse embryonic fibroblasts that express just the MH2 domain of SMAD2 (MEF SMAD2$^{\Delta ex2}$) (*Das et al., 2009*), treated with 2 ng/ml TGF-β. The untreated sample is only shown for the WT SKI peptide. A PSMAD2 immunoblot is shown. (**D**) A recombinant trimer of phosphorylated SMAD2 MH2 domain was used in a peptide pulldown assay with WT and G34D SKI peptides. A PSMAD2 immunoblot is shown, with inputs on the right. (**E**) Mutational peptide array of SKI peptides (amino acids 11–45), mutated at all residues between amino acids 19 and 35, was probed with a recombinant PSMAD3–SMAD4 complex, which was visualized using a SMAD2/3 antibody conjugated to Alexa 488. On each row, the indicated amino acid is substituted for every other amino acid. A representative example is shown. See *Figure 4—figure supplement 1C* and *Figure 4—source data 2* for quantification of the peptide arrays.

The online version of this article includes the following source data and figure supplement(s) for figure 4:

**Source data 1.** Peptide sequences for peptide array.
**Source data 2.** Quantification of peptide arrays.
**Figure supplement 1.** SGS mutations in SKI.

---

a pseudo-phosphorylated SMAD2 MH2 domain produced in *Escherichia coli* was used, complexed with a SKI peptide containing a C-terminal acidic tag (Ser-Asp-Glu-Asp).

Since our structure was generated with phosphorylated SMAD2, we were able to explore why SKI only binds phosphorylated R-SMADs and not monomeric unphosphorylated R-SMADs. To do this we compared the structure of the unphosphorylated SMAD2 MH2 domain bound to a region of ZFYVE9 (formerly called SARA) (*Wu et al., 2000*) with our current structure of phosphorylated SMAD2 MH2 domain complexed with SKI. It was clear that in the unphosphorylated SMAD2 structure, Tyr268 in the so-called β1' strand (amino acids 261–274) is locked in a stable conformation in a hydrophobic pocket, and also forms a number of hydrogen bonds (*Figure 5G*). Crucially, this conformation forces Trp448 into flattened orientation, which is incompatible with SKI binding through the pi-stacking involving SKI Phe24, SMAD2 Trp448, and SKI Pro35 (*Figure 5G*). MH2 domain trimerization generates a new binding site for the β1' strand on the adjacent MH2 domain subunit (*Figure 5H*; *Video 1*). The central residue driving this is Tyr268. In the trimer, the hydroxy group of Tyr268 makes hydrogen bond contact with the carbonyl group of Asp450 and the main chain of Lys451 on the adjacent MH2 domain subunit. As a consequence, Trp448 moves into an upright position in the trimer, allowing engagement with SKI. Thus, SKI can only bind SMAD2 in its phosphorylated trimeric state.

## Knockin of an SGS mutation into HEK293T cells inhibits Activin-induced SKI degradation and attenuates PSMAD3–SMAD4-mediated transcriptional activity

We have shown that the presence of SGS mutations prevent the interaction of SKI/SKIL with phosphorylated R-SMADs and have demonstrated that SKI/SKIL degradation requires an activated R-SMAD–SMAD4 complex. We therefore went on to investigate the functional effect of the SGS mutations on SKI degradation and TGF-β/Activin-induced transcriptional responses. To do this, we chose to focus on Pro35 because of its crucial role in forming the stacking interaction with Trp448 in SMAD2. In SGS patients, Pro35 is mutated to Ser or Gln (*Carmignac et al., 2012*; *Doyle et al., 2012*; *Schepers et al., 2015*), substitutions not tolerated in activated R-SMAD–SMAD4 binding (*Figure 4E*).

We used CRISPR/Cas9 technology with a single-stranded template oligonucleotide to knock in the Pro35 → Ser (P35S) mutation into HEK293T cells, and we efficiently generated a number of homozygous clones (*Figure 6—figure supplement 1A*). In three independent clonal cell lines carrying the P35S SKI mutation, the binding to endogenous phosphorylated SMAD2 was severely compromised, compared with WT SKI (*Figure 6A*). The binding to SMAD4, however, was unchanged in the mutant cell lines, as the SGS mutations do not affect the SKI SAND domain, which is responsible for SMAD4 binding (*Figure 6A*). To assess the impact of the P35S SKI mutation on the SKI and SKIL degradation, cells were treated with Activin for 1 or 2 hr and SKI/SKIL levels determined by immunoblotting. At both time points, we clearly demonstrated that P35S SKI levels remained stable, whilst in the parental cell lines, SKI protein is almost entirely degraded after 1 hr of Activin treatment (*Figure 6B*). The presence of mutated SKI had no effect on the Activin-induced degradation of SKIL in these lines (*Figure 6B*). Thus, the P35S mutation renders SKI completely resistant to ligand-induced degradation.

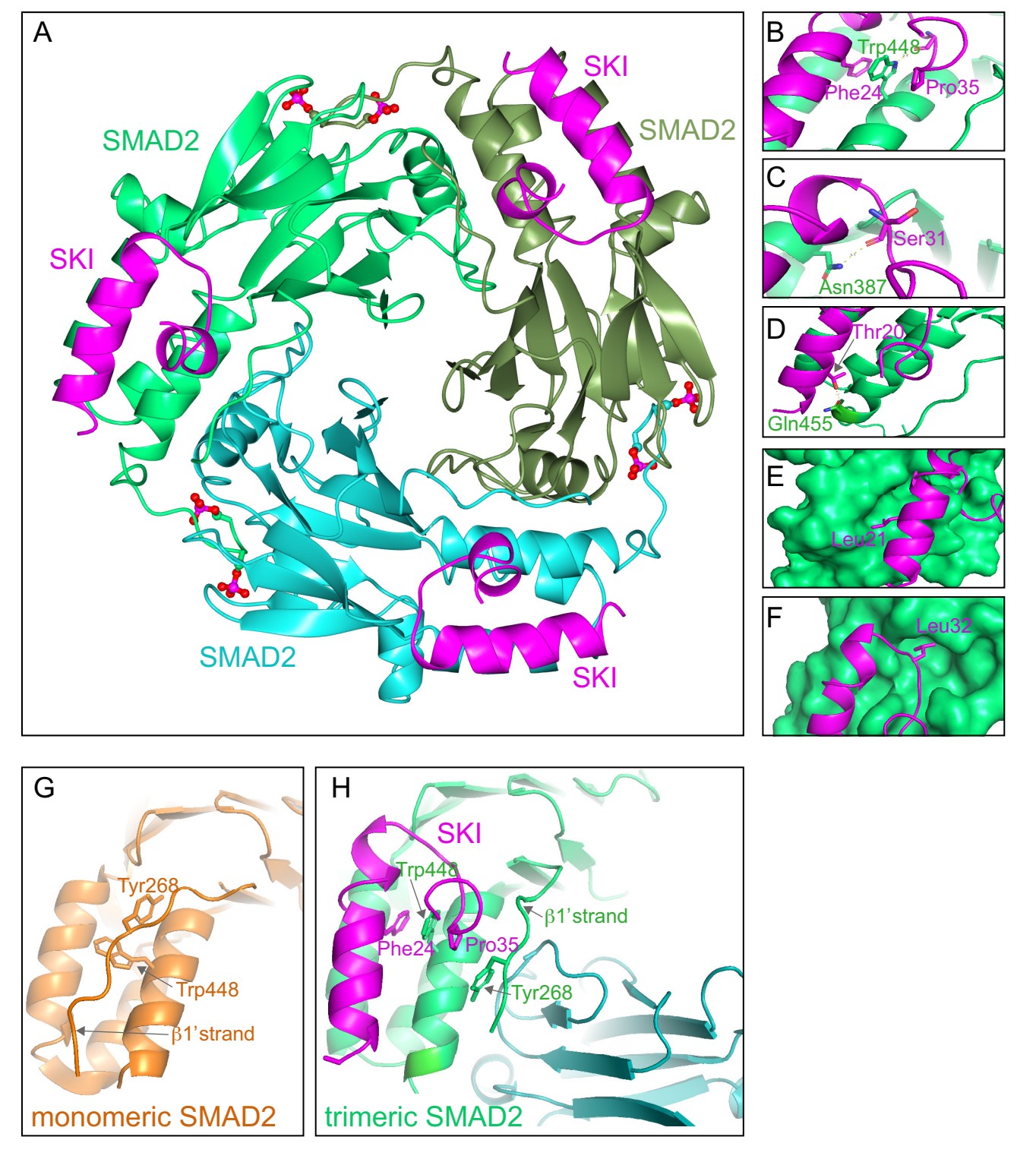

**Figure 5.** Crystal structure of PSMAD2 MH2 domain and N-terminal SKI peptide. (**A**) Crystal structure of the phosphorylated SMAD2 MH2 domain trimer (the three monomers are shown in bright green, cyan, and olive) with the N-terminal SKI peptide amino acids 11–45 (magenta). A ribbon representation is shown. The C-terminal phosphates are indicated with a ball and stick representation (red and magenta). (**B–F**) Close ups on key residues for SKI binding. SKI residues are shown in magenta, and SMAD2 residues are in green. In (**B–D**), a ribbon representation is shown. In (**E and F**), SMAD2 is shown as a surface representation and SKI as a ribbon. (**G**) A detail from the structure of monomeric SMAD2 MH2 domain with a peptide from ZFYVE9 (formerly called SARA) (**Wu et al., 2000**). Note that the β1' strand that contains Tyr268 is locked in a hydrophobic pocket, forcing Trp448

*Figure 5 continued on next page*

*Figure 5 continued*

into flattened orientation, incompatible with SKI binding. (**H**) A detail from the structure in (**A**) indicating how SMAD2 complex formation shifts the position of the β1' strand and more particularly, Tyr268, allowing Trp448 to flip 90°, enabling it to stack with SKI residues Phe24 and Pro35.

The online version of this article includes the following source data and figure supplement(s) for figure 5:

**Source data 1.** Structure validation report for crystal structure (ID: 6ZVQ).

**Figure supplement 1.** Analysis of the phosphorylated SMAD2 MH2 domain complex used for structural studies.

To determine whether SGS mutations had the same effect in SKIL, we introduced a G103V mutation into SKIL, corresponding to the SGS mutation G34V in SKI (referred to as SKIL ΔS2/3). Transfection of G103V SKIL in HEK293T cells led to reduction of SMAD2 binding in parental cells (*Figure 6—figure supplement 1B*). The residual binding was mediated via SMAD2's interaction with SMAD4, as it was lost in the SMAD4 knockout cells (*Figure 6—figure supplement 1B*). Binding of SMAD4 in the absence or presence of signal was unaffected by the mutation. As observed above for SKI, the SGS mutation in SKIL led to resistance to Activin-induced degradation (*Figure 6—figure supplement 1C*), indicating that the R-SMAD interaction was essential. In addition, we made a version of SKIL with mutations in the SAND domain (R314A, T315A, H317A, and W318E) that rendered it unable to interact with SMAD4 (referred to as SKIL ΔS4). This mutant was also not degraded upon Activin stimulation (*Figure 6—figure supplement 1C*), demonstrating an essential requirement for SMAD4 binding.

SKI and SKIL bind DNA in conjunction with SMAD4 at SBEs of TGF-β/Activin target genes in the absence of ligand stimulation. The ligand-induced degradation of SKI and SKIL then allows the activated R-SMAD–SMAD4 complexes access to the SBEs to activate transcription of target genes (*Levy et al., 2007*; *Stroschein et al., 1999*). We hypothesized that if the SGS mutations render SKI resistant to ligand-induced degradation, then mutant SKI and SMAD4 would remain bound to the DNA. To test this, we used a DNA pulldown assay with an oligonucleotide corresponding to SBEs from the *JUN* promoter (*Levy et al., 2007*). Consistent with our prediction, both WT and P35S SKI bound the SBEs with SMAD4 in the absence of signal, but after Activin stimulation, the binding of WT SKI is lost, whilst the binding of P35S SKI is retained (*Figure 6C*).

The SKI–SMAD4 complex bound to SBEs in the absence of signal are transcriptionally repressive (*Levy et al., 2007*; *Stroschein et al., 1999*). Since P35S SKI remains bound with SMAD4 in Activin-stimulated cells, we reasoned that this would inhibit Activin-induced gene expression. To address this, we stably expressed luciferase reporters (CAGA$_{12}$-Luciferase and BRE-Luciferase), together with TK-Renilla as an internal control (*Dennler et al., 1998*; *Korchynskyi and ten Dijke, 2002*), in parental HEK293T cells and in two independent clones of the knockin P35S SKI cells. The CAGA$_{12}$-Luciferase reporter responds to TGF-β and Activin, is induced by PSMAD3–SMAD4 complexes, and is sensitive to SKI and SKIL levels, whilst the BRE-Luciferase reporter is induced by SMAD1/5–SMAD4 complexes in response to BMPs and is not affected by SKI and SKIL (*Levy et al., 2007*). Strikingly, we found a significant reduction in Activin-induced CAGA$_{12}$-Luciferase activity in the P35S SKI cells compared to the parental cell line (*Figure 6D*), while BMP4-induced BRE-Luciferase activity was similar in all cell lines (*Figure 6E*). The results indicate that SGS mutations in SKI lead to inhibition of TGF-β/Activin-induced transcription mediated by PSMAD3–SMAD4 complexes.

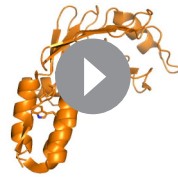

**Video 1.** Mechanism of SKI binding to phosphorylated SMAD2 MH2 domain. Animation of SMAD2 MH2 domain monomer (orange) forming a complex with two other SMAD2 MH2 domain monomers (cyan and olive). Note the movement of Trp448 on helix five and Tyr268 on the β1' strand in the orange monomer upon trimerization. The flipped Trp448 is then in the correct orientation for binding to the SKI peptide (magenta).
https://elifesciences.org/articles/63545#video1

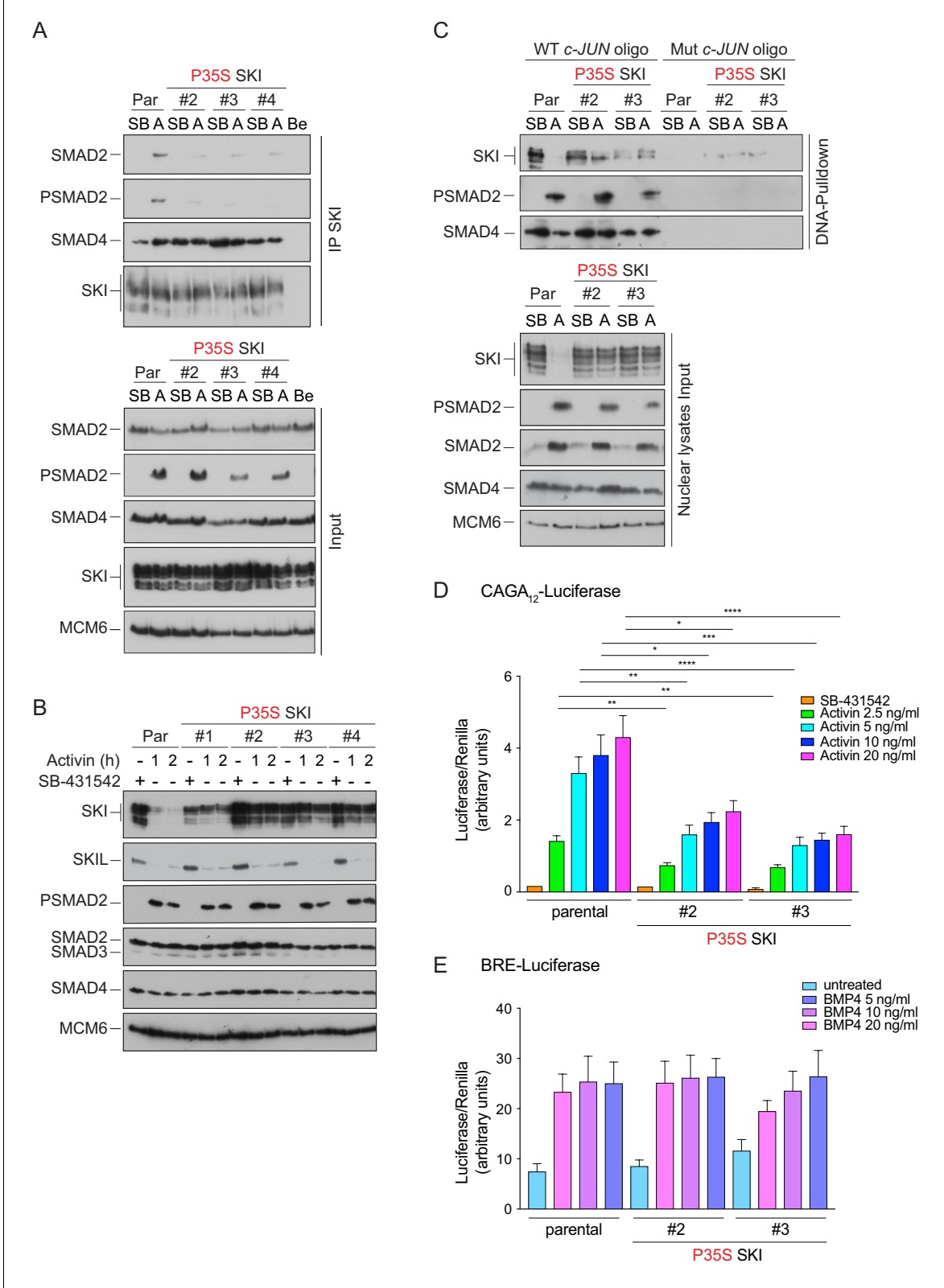

**Figure 6.** Knockin of an SGS mutation into SKI in HEK293T cells inhibits SKI degradation and inhibits Activin-induced transcription. (A) Parental HEK293T and three independent P35S SKI knockin clones were incubated overnight with 10 μM SB-431542, washed out, and treated for 3 hr with 25 μM MG-132 and then with either SB-431542 or 20 ng/ml Activin A for an additional 1 hr. Whole-cell lysates were immunoprecipitated (IP) with SKI antibody or beads alone (Be). The IPs were immunoblotted using the antibodies shown. Inputs are shown below. (B) Parental HEK293T and four independent

*Figure 6 continued on next page*

*Figure 6 continued*

P35S SKI knockin clones were incubated with 10 µM SB-431542 overnight, washed out, and incubated with either SB-431542 or 20 ng/ml Activin for the times indicated. Whole-cell lysates were immunoblotted using the antibodies indicated. (C) Cells were treated as in (B), and nuclear lysates were prepared and analyzed by DNA pulldown assay using the wild-type c-Jun SBE oligonucleotide or a version mutated at the SMAD3–SMAD4 binding sites (top panel). Inputs are shown in the bottom panel. HEK293T parental and two independent P35S SKI knockin clones were stably transfected with the CAGA$_{12}$-Luciferase reporters (D) or the BRE-Luciferase reporter (E) with TK-Renilla as an internal control. Cells were serum starved with media containing 0.5% fetal bovine serum and 10 µM SB-431542 overnight. Subsequently, cells were washed and treated with Activin A (D) or BMP4 (E) at the concentrations indicated for 8 hr. Cell lysates were prepared and assayed for Luciferase and Renilla activity. Plotted are the means and SEM of seven (D) or four (E) independent experiments, with the ratio of Luciferase:Renilla shown. *p<0.05; **p<0.01; ***p<0.001; ****p<0.0001. The p-values are from two-way ANOVA with Tukey's post hoc test. A, Activin; SB, SB-431542; Par, parental.

The online version of this article includes the following source data and figure supplement(s) for figure 6:

Source data 1. Luciferase assays for Activin A-induced HEK293T P35S SKI clones, as presented in *Figure 6D*.
Source data 2. Luciferase assays for BMP4-induced HEK293T P35S SKI clones, as presented in *Figure 6E*.
Figure supplement 1. Mutation of the R-SMAD binding domain or SAND domain in SKI/SKIL prevents ligand-induced SKI/SKIL degradation.

## Dermal fibroblasts from SGS patients exhibit an attenuated transcriptional TGF-β response

To gain further insights into the functional consequences of the SGS mutations, we obtained dermal fibroblasts from two SGS patients: a female carrying the heterozygous point mutation L32V and a male carrying a heterozygous deletion of 12 base pairs corresponding to codons 94–97 (ΔS94-97) (*Carmignac et al., 2012*). Both patients present the classical features of SGS such as marfanoid habitus and intellectual disability, whilst the patient-carrying L32V mutation also manifests craniosynostosis. In addition, we obtained dermal fibroblast from a healthy male subject as a control.

We investigated whether the SGS mutations rendered SKI resistant to TGF-β-induced degradation in the dermal fibroblasts, as demonstrated in the knockin HEK293T cells. Indeed, in control fibroblasts, SKI expression is abrogated upon TGF-β stimulation after 1 hr, while in the SGS-derived fibroblasts, SKI protein remains relatively stable (*Figure 7A*). Note that these cells are heterozygous for the mutation, while the HEK293T P35S SKI knockins used above were homozygous, thus accounting for the incomplete SKI stabilization exhibited by the SGS fibroblasts, compared with the knockin cells.

To determine the effect of SKI stabilization on global TGF-β-induced transcription we performed genome-wide RNA-sequencing (RNA-seq) in three different conditions: SB-431542-treated cells (non-signaling condition), 1 hr TGF-β-treated and 8 hr TGF-β-treated, and compared the L32V and ΔS94-97 SKI fibroblasts to control fibroblasts. The samples separated in a principal component analysis according to the cell line used and the treatment performed (*Figure 7—figure supplement 1A*), and we confirmed that the differentially enriched genes after TGF-β treatment in the control fibroblasts were characteristic of pathways related to TGF-β signaling (*Figure 7—figure supplement 1B*). We then performed a pairwise comparison between the TGF-β-treated and SB-431542-treated samples for each of the cell lines individually. We found that of the 339 genes that were differentially expressed in normal fibroblasts after 1 hr of TGF-β treatment, 60% (202 genes) were induced or repressed less efficiently in the L32V mutant fibroblasts, and of these, 97 genes were not significantly differentially expressed in the mutant cells at all (*Figure 7—source data 1*, *Figure 7B,C*). After an 8 hr TGF-β induction, we found that 4769 genes were differentially expressed in the normal fibroblasts, and of these 75% (3556 genes) were induced or repressed less efficiently in the L32V mutant cells, and 880 genes were not significantly differentially expressed in the mutant cells at all (*Figure 7—source data 1*, *Figure 7B,C*). We observed similar results when comparing the TGF-β responses in the normal fibroblasts versus ΔS94-97 SKI fibroblasts, although the effects were less dramatic (*Figure 7—source data 1*, *Figure 7—figure supplement 1C,D*). To illustrate the magnitude of these effects we validated six gene expression profiles (*ISLR2, CALB2, SOX11, ITGB6, HEY1, COL7A1*) in the normal versus mutant fibroblasts by qPCR (*Figure 7—figure supplement 2*).

Thus, we conclude that the presence of SGS point mutations in SKI that render it resistant to ligand-induced degradation, result in attenuated TGF-β responses for a substantial subset of target genes.

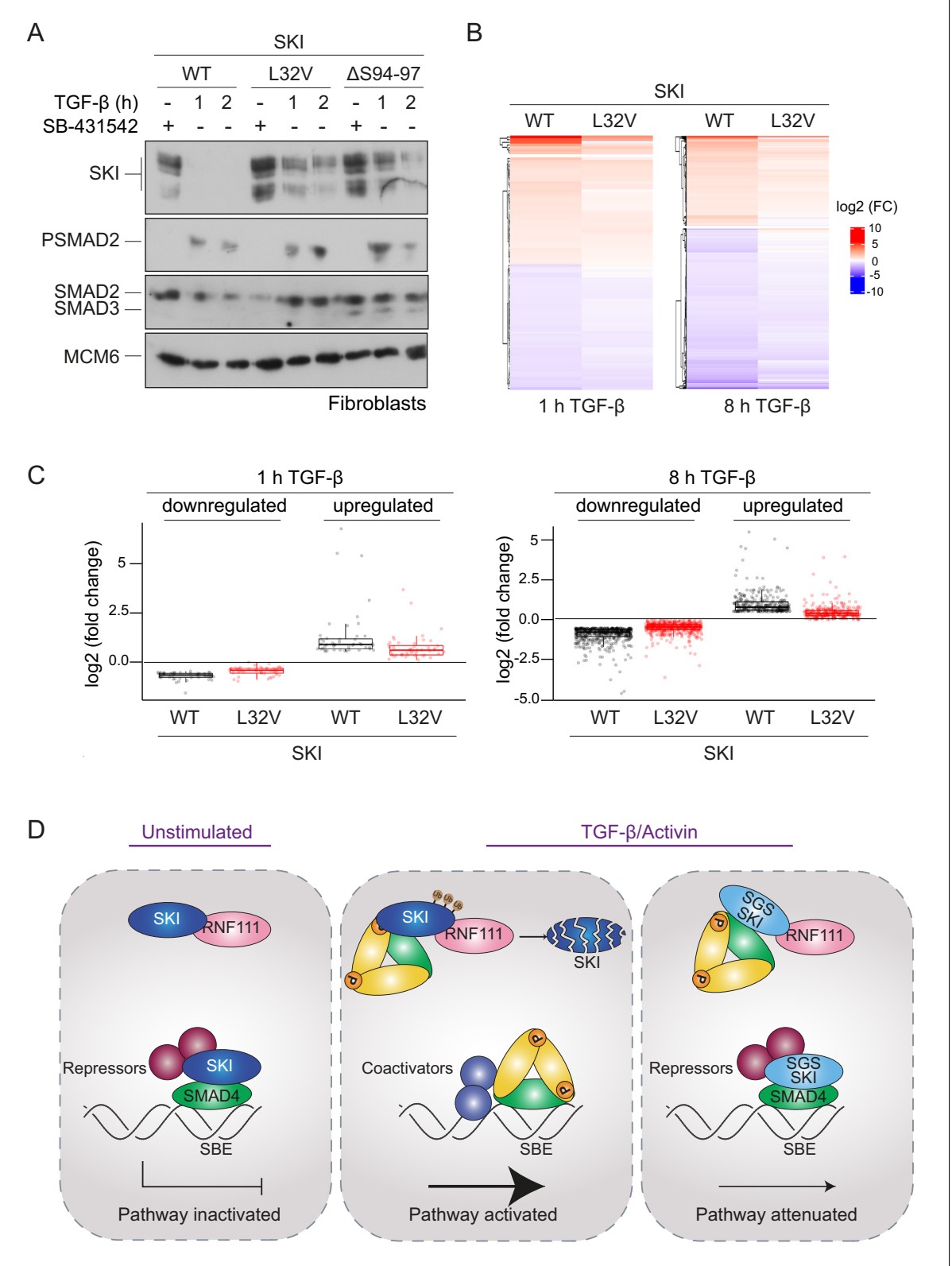

**Figure 7.** SGS mutations in SKI inhibit TGF-β-induced transcriptional responses in fibroblasts derived from SGS patients. (**A**) Fibroblasts derived from a healthy subject carrying WT SKI and from two SGS patients carrying the L32V or the ΔS94-97 heterozygous mutations in SKI were incubated overnight with 10 μM SB-431542, washed out, and either re-incubated with SB-431542 or 2 ng/ml TGF-β for the times indicated. Whole-cell lysates were immunoblotted using the antibodies indicated. (**B**) Hierarchically clustered heatmaps of log2FC values (relative to the SB-431542-treated samples)

*Figure 7 continued on next page*

*Figure 7 continued*

showing the expression of TGF-β-responsive genes in the healthy fibroblasts and the L32V SKI fibroblasts after 1 hr and 8 hr of TGF-β treatment, analyzed by RNA-seq. Four biological replicates per condition were analyzed. The genes shown are those for which the TGF-β inductions were statistically significant in the healthy fibroblasts, but non-significant in the L32V fibroblasts. (C) The same data as in (B) are presented as box plots. (D) Model for the mechanism of action of WT SKI and mutated SKI. The left panel shows the unstimulated condition. In the nuclei, SKI (blue) is complexed with RNF111 (pink) and is also bound to DNA at SBEs with SMAD4 (green) forming a transcriptionally repressive complex with other transcriptional repressors (maroon). In the middle panel, TGF-β/Activin stimulation induces the formation of phosphorylated R-SMAD–SMAD4 complexes (yellow and green), which induce WT SKI degradation by RNF111. This allows an active PSMAD3–SMAD4 complex to bind SBEs and activate transcription. In the right panel, SGS-mutated SKI (light blue) is not degraded upon TGF-β/Activin stimulation, due to its inability to interact with PSMAD2 or PSMAD3. It therefore remains bound to SMAD4 on DNA, leading to attenuated transcriptional responses.

The online version of this article includes the following source data and figure supplement(s) for figure 7:

**Source data 1.** RNA-seq raw data.
**Figure supplement 1.** Dermal fibroblasts from SGS patients exhibit an attenuated TGF-β transcriptional response.
**Figure supplement 2.** Validation of RNA-seq data by qPCR.
**Figure supplement 2—source data 1.** qPCR validations of RNA-seq data for SGS and control dermal fibroblasts.

## Discussion

### SGS mutations in SKI lead to stabilization of SKI and attenuated TGF-β/Activin transcriptional responses

In this study, we have resolved the mechanism of action of SKI and the related protein SKIL, which has allowed us to elucidate the molecular consequences of SKI mutations in SGS. Our proposed mechanism of how SKI acts as a transcriptional repressor of TGF-β/Activin signaling in health and disease is illustrated in *Figure 7D*, and our results suggest that the mechanism of action of SKIL is equivalent. In the absence of ligand stimulation, in both healthy and diseased cells, SKI and SKIL bind in conjunction with SMAD4 at SBEs of TGF-β/Activin target genes. Here, they repress transcription by recruiting corepressors such as NCOR1 or SIN3A (*Tokitou et al., 1999*; *Nomura et al., 1999*; *Stroschein et al., 1999*; *Deheuninck and Luo, 2009*). We and others have also shown that in unstimulated cells, SKI and SKIL interact with the E3 ubiquitin ligase, RNF111, although this binding per se does not lead to SKI/SKIL degradation (*Le Scolan et al., 2008*; *Levy et al., 2007*; *Nagano et al., 2007*). In healthy cells, upon TGF-β/Activin stimulation, SKI/SKIL form a complex with a canonical PSMAD2/PSMAD3–SMAD4 trimer, which induces rapid degradation of SKI/SKIL via RNF111 (*Le Scolan et al., 2008*; *Levy et al., 2007*; *Nagano et al., 2007*). A possible mechanism explaining this would be that the binding of the PSMAD2/PSMAD3–SMAD4 complex induces an activating conformational change in RNF111, although this has not yet been demonstrated. Degradation of SKI and SKIL removes the repressors from the SBEs, allowing access of activated PSMAD3–SMAD4 complexes to the SBEs to regulate transcription of target genes. In SGS cells, mutated SKI can no longer interact with PSMAD2/PSMAD3, and it is therefore not degraded upon TGF-β/Activin signaling. It thus remains bound with SMAD4 to SBEs, resulting in an attenuation of TGF-β/Activin transcriptional responses. To demonstrate the functional consequences of the SGS mutations, we used genome-wide RNA-seq analysis of fibroblasts derived from SGS patients and have shown that SGS mutations indeed lead to a reduction in the magnitude of TGF-β transcriptional responses.

### The mechanism underlying SKI and SKIL function

Since the discovery that SKI and SKIL interact with SMAD2 and SMAD3, and act as negative regulators of TGF-β/Activin pathways (*Luo et al., 1999*; *Stroschein et al., 1999*; *Sun et al., 1999*), two different mechanisms of action have been proposed. One mechanism is as described in the paragraph above – an initial version of which was first proposed in 1999 (*Stroschein et al., 1999*). The second mechanism was based on the crystal structure of the SKI SAND domain with the MH2 domain of SMAD4 (*Wu et al., 2002*). In this crystal structure, the binding of SKI with SMAD4, which is mediated via the I-loop of the SKI SAND domain with the L3 loop of SMAD4, was mutually exclusive with the binding of SMAD4 to activated R-SMADs, which also requires the L3 loop of SMAD4. Thus, the authors concluded that the mechanism whereby SKI (and by analogy, SKIL) inhibited TGF-β/Activin signaling was by binding to the activated R-SMADs and SMAD4 in such a way as to disrupt the

phosphorylated R-SMAD–SMAD4 complexes required for transcriptional activation (*Wu et al., 2002*). This mechanism has been supported by the observation that overexpression of SKI and SKIL inhibits TGF-β-induced functional responses (*Luo, 2004*). The two mechanisms are fundamentally different. In the first, SKI and SKIL are constitutive repressors that need to be degraded to allow pathway activation. In the second, SKI and SKIL act as inducible repressors, as they repress only upon ligand induction, by virtue of their ability to disrupt activated SMAD complexes.

Our biochemical analysis of the role of SMAD4 in SKI/SKIL function now resolves the controversy between the two models. First, the efficient SKIL and SKI degradation that we and others have observed upon ligand stimulation in effect rules out the second model, at least in the first hours after ligand induction, as there would be little or no nuclear SKI or SKIL to disrupt activated SMAD complexes. Second, we clearly demonstrate for the first time that an intact functional phosphorylated R-SMAD–SMAD4 trimer is required to bind to SKIL to induce its ligand-dependent degradation. The key piece of evidence for this comes from our analysis of the inability of SMAD4 point mutants to restore ligand-induced SKIL degradation in SMAD4-null HaCaTs. Critically, we show that SMAD4 mutants that cannot form a canonical activated R-SMAD–SMAD4 trimer cannot rescue ligand-induced SKIL degradation, neither can SMAD4 mutants that do not interact with the SAND domain of SKIL. Thus, we conclude that TGF-β/Activin-induced SKIL degradation occurs only when SKIL interacts simultaneously with phosphorylated R-SMADs and with SMAD4, which in turn must interact with each other in a transcriptionally active trimer. This strongly indicates that SKIL binding to the R-SMAD–SMAD4 complex does not disrupt it.

This conclusion is supported by a recent crystal structure of the SAND domain of SKIL with the SMAD4 MH2 domain (*Walldén et al., 2017*). This structure revealed that SKIL interacts with SMAD4 in two states: an 'open' and a 'closed' conformation. In the open conformation, the authors showed that SKIL can bind the R-SMAD–SMAD4 complex without intermolecular clashes or further structural readjustment, whereas in the closed state, structural reorganization within the SMAD heterotrimer is required to allow binding of SKIL, as has been observed in the previous structure of SKI with SMAD4 (*Wu et al., 2002*). Molecular modelling has subsequently confirmed that SKIL in the open conformation forms a stable ternary SKIL–SMAD3–SMAD4 complex (*Ji et al., 2019*). Furthermore, surface plasmon resonance indicated only one dominant binding mode for SKIL and SMAD4, leading to the conclusion that the open conformation is the biologically and functionally relevant mode and that the closed conformation may be the result of crystal packing forces (*Walldén et al., 2017*). Note that the residues that allow the binding in the open conformation are highly conserved between SKI and SKIL, suggesting that both repressors bind to an intact activated R-SMAD–SMAD4 complex, which is required for their degradation. This would exclude the disruption model and thus favors the degradation model. It will now be important to solve the structure of an activated R-SMAD–SMAD4 trimer with the N-terminal half of SKI or SKIL that contains the R-SMAD binding motif, the DHD domain and the SAND domain that contacts the SMAD4 moiety. The role of the DHD domain is particularly intriguing as SGS mutations also occur in this domain (*Carmignac et al., 2012*; *Doyle et al., 2012*; *Schepers et al., 2015*) and appear from our patient sample analysis to have a similar effect on inhibiting ligand-induced SKI degradation.

Our structural data also elegantly explain why SKI and SKIL only bind to phosphorylated SMAD2 and SMAD3 in the context of an activated SMAD trimer, and not to monomeric SMAD2 or SMAD3. The key residue for this discrimination is Trp448 in SMAD2, the equivalent of which would be Trp406 in SMAD3. In the trimer, this residue is in a conformation compatible with stacking between Phe24 and Pro35 of SKI. In the monomer, however, it is rotated approximately 90°, prohibiting SKI binding. The binding mode of SKI to SMAD2 is distinct from that of other SMAD2-binding partners, for example, the transcription factor FOXH1, which contains two binding motifs (*Miyazono et al., 2018*). One of which, the so-called SIM, binds SMAD2 in both monomeric and trimeric forms, whilst the so-called FM only binds phosphorylated trimeric SMAD2 because it recognizes the interface of the SMAD trimer (*Miyazono et al., 2018*; *Randall et al., 2002*; *Randall et al., 2004*). It will be interesting in the future to discover whether any other SMAD2/3-binding partner uses the same mode of interaction as SKI and SKIL.

## The molecular mechanism of SGS

As discussed in Introduction, SGS is a Marfan-related syndrome, with patients exhibiting many of the same features characteristic of MFS and LDS. These syndromes have been considered as TGF-β

signalopathies, as the causal mutations are either direct components of the TGF-β signaling pathway or, as in the case of MFS, a component of the microfibrils in the ECM, FBN1, that is known to bind latent TGF-β in complex with latent TGF-β binding proteins (LTBPs) (*Cannaerts et al., 2015*; *Ramirez et al., 2004*; *Robertson et al., 2015*). There has been controversy over whether the manifestations of these syndromes result from too little TGF-β signaling, or too much. This is obviously a crucial issue to resolve, as it is influencing the types of treatments being developed for patients with these syndromes.

The first suggestion that MFS resulted from excessive TGF-β came from mouse models, where key phenotypes could be rescued by a TGF-β neutralizing antibody (*Habashi et al., 2006*; *Neptune et al., 2003*). However, later studies using a potent murine anti-TGF-β antibody, or genetic methods for reducing TGF-β signaling have not corroborated these findings, and have worsened, rather than improved disease in MFS mouse models (*Cook et al., 2015b*; *Holm et al., 2011*; *Lindsay et al., 2012*; *Wei et al., 2017*). Furthermore, administration of small-molecule inhibitors of the TGF-β type I receptor, or a pan-TGF-β neutralizing antibody, has been associated with serious adverse cardiovascular toxicities, such as valve defects, similar to those found in MFS (*Anderton et al., 2011*; *Stauber et al., 2014*; *Mitra et al., 2020*). The finding that mutations in FBN1 that prevent binding of LTBPs might result in lower levels of TGF-β signaling, rather than excessive signaling, is not so surprising given the more recent understanding of how TGF-β is activated. In order for mature TGF-β ligands to be released from the latent complex, either force has to be applied via integrins, to partially unfold the cleaved TGF-β pro-domain allowing release of the mature domain, or the pro-domain must be degraded by proteases (*Dong et al., 2017*; *Rifkin et al., 2018*; *Robertson and Rifkin, 2016*). For the traction mechanism to occur, the integrin must be anchored to the actin cytoskeleton, and the LTBP must be tethered to ECM, via FBN1 and fibronectin. Thus, release of latent TGF-β alone is not therefore sufficient to produce active TGF-β ligands.

Consistent with the view that lower levels of TGF-β signaling might be responsible for MFS, the mutations that give rise to LDS are all loss-of-function mutations in TGF-β pathway components (*Schepers et al., 2018*). Paradoxically though, signatures of higher TGF-β signaling were observed over time in mouse models of LDS (*Gallo et al., 2014*; *MacFarlane et al., 2019*). However, in this case, the pathology could not be rescued by neutralizing TGF-β activity. One possibility with both the mouse models of MFS and LDS is that the mutations do initially lead to lowered TGF-β signaling, but over time cells compensate by up-regulating either TGF-β ligands themselves or other TGF-β family ligands that signal through PSMAD2/PSMAD3, ultimately leading to the enhanced signaling signatures observed.

With respect to SGS, there has been much less research into the consequences of the SGS mutations on TGF-β signaling responses. Based on the SMAD complex disruption model of SKI/SKIL action, it has been assumed that loss-of-function mutations in a negative regulator would lead to an increase in TGF-β signaling, and in fact, this has been used to support the idea that the Marfan-related syndromes are caused by excessive TGF-β signaling (*Doyle et al., 2012*; *Gallo et al., 2014*). Here we unequivocally show that the opposite is true. We demonstrate that these mutations lead to loss of ligand-induced SKI degradation. As a result, the stabilized SKI remains bound to SBEs with SMAD4 as a repressive complex, and hence, a subset of TGF-β/Activin transcriptional responses are attenuated. We have proven this in both HEK293T knockin cells and patient-derived fibroblasts. Moreover, we also find no evidence for increased PSMAD2 or PSMAD3 signaling. Indeed, neither model of SKI function would actually predict that the SKI mutations would affect levels of phosphorylated R-SMADs, since SKI acts downstream of R-SMAD phosphorylation. Finally, our finding that SGS mutations in SKI lead to its stabilization and are not equivalent to loss of SKI function also explains why patients with 1p36 deletion syndrome, who are haploinsufficient for SKI, do not have SGS. However, unsurprisingly, many of the same organs are affected in both syndromes (*Colmenares et al., 2002*; *Zhu et al., 2013*).

It will now be important to use animal models to explore how attenuation of transcription of specific TGF-β/Activin target genes leads to the manifestations of SGS, and to understand why SGS patients exhibit additional defects compared with LDS and MFS patients. We anticipate that our new understanding that the SKI mutations lead to attenuation of TGF-β responses will resolve the paradoxes surrounding the role of aberrant TGF-β signaling in the other Marfan-related disorders and will help inform the development of new therapeutic approaches.

## Materials and methods

### Cell lines

HEK293T and HaCaT cells were obtained from the Francis Crick Institute Cell Services and cultured in Dulbecco's modified Eagle's medium (DMEM) supplemented with 10% fetal bovine serum (FBS) and 1% Penicillin/Streptomycin (Pen/Strep). All CRISPR-Cas9 edited cell lines were cultured in the same media. Dermal fibroblasts from healthy subjects were kindly provided by David Abraham (UCL-Medical School Royal Free Campus) under the ethics of the Health Research Authority, NRES Committee London – Hampstead, Research Ethics Committee (REC) reference, 6398. L32V and ΔS94-97 SKI dermal fibroblasts were obtained from Laurence Faivre and Virginie Carmignac (Université de Bourgogne UMR1231 GAD, Dijon, France) under the ethics of the GAD collection, number DC2011-1332 (*Carmignac et al., 2012*). The mutations were confirmed by Sanger sequencing and RNA sequencing. The fibroblasts were all cultured in DMEM supplemented with 10% FBS, 1% Pen/ Strep, and 1% insulin–transferrin–selenium (Thermo Fisher). Mouse embryo-derived fibroblasts harboring the homozygous null allele $Smad2^{ex2}$ (MEF SMAD2$^{ex2}$) (*Piek et al., 2001*) were maintained in DMEM supplemented with 10% FBS and 1% Pen/Strep. All cell lines have been banked by the Francis Crick Institute Cell Services and certified negative for mycoplasma. The identity of all cell lines was also authenticated by confirming that their responses to ligands and their phenotypes were consistent with published history. All the cell lines are listed in Key Resources Table.

### Ligands, chemicals, and cell treatments

Ligands and inhibitors were used at the following concentrations: TGF-β (PeproTech), 2 ng/ml; Activin A (PeproTech), 20 ng/ml; BMP4 (PeproTech), 20 ng/ml; SB-431542 (Tocris), 10 µM; MG-132 (Tocris), 25 µM. All treatments were performed in full serum or, where required, in serum-starved (0.5% FBS) DMEM. Unless otherwise stated, cells were incubated with 10 µM SB-431542 overnight to inhibit autocrine signalling, then were washed three times with warm media, and stimulated with either Activin A or TGF-β. For proteasome inhibition, cells were treated for 3 hr with 25 µM MG-132 prior stimulation with Activin A or TGF-β.

### Plasmids

Plasmids are listed in Key Resources Table. CAGA$_{12}$-Luciferase, BRE-Luciferase, TK-Renilla, pEGFP-C1, and pEGFP-SMAD4 were as described previously (*Dennler et al., 1998*; *Korchynskyi and ten Dijke, 2002*; *Levy et al., 2007*; *Nicolás et al., 2004*). The pEGFP-SMAD4 mutants (D351H and D537Y) were generated by swapping the mutated SMAD4 coding region from the EF-HA vector into pEGFP-C1 (*De Bosscher et al., 2004*). The pEGFP-SMAD4 SMAD4 mutants (A433E and I435Y) were generated by PCR using oligonucleotides listed in Key Resources Table. EF-Flag-SKIL G103V was generated from pEF-FLAG-SKIL by PCR using oligonucleotides listed in Key Resources Table, while the mutant containing the R314A, T315A, H317A, and W318E mutations was generated by synthesizing the SKIL region between BSTEII and AVRII sites containing the mutations and cloning that fragment into pEF-FLAG-SKIL. For plasmids used for generating recombinant proteins, see below.

### Transfections, generation of stable cell lines, and reporter assays

Cells were transfected with the appropriate plasmids using Fugene 6 (Roche) according to the manufacturer's instructions. Luciferase reporter assays were performed as previously described, using the Dual-Glo assay system (Promega) following the manufacturer's instructions (*Levy et al., 2007*).

HaCaT SMAD4 KO lines stably expressing either EGFP or EGFP-SMAD4 WT or mutants were generated by transfecting the cells with the appropriate plasmids. Transfected cells were selected with 500 µg/ml of G418 (Invitrogen), then FACS sorted for EGFP-positive cells, and expanded. EGFP expression was confirmed by microscopy. To generate stable HEK293T cell lines expressing either CAGA$_{12}$-Luciferase or BRE-Luciferase together with TK-Renilla, cells were transfected with the appropriate plasmids together with a plasmid carrying the puromycin resistance gene (pSUPER-retro-puro; OligoEngine). Cells were then selected with 2 µg/ml puromycin (Sigma).

## CRISPR/Cas9-mediated knockout of SMADs in HEK293T and HaCaT cells

For the generation of knockin or knockout HEK293T cells, a parental clone was selected, which was a representative clone from the HEK293T pool that showed a robust Activin-induced SKI degradation and responded to TGF-β family ligands in the same way as the starting pool. For HaCaTs, a pool of cells was used as starting material for knockouts.

For SMAD2 and SMAD3 knockouts, a guide RNA in the MH2 domain of the protein was selected, whereas for SMAD4, two guide RNAs were picked, one targeting the MH1 domain and the other targeting at the end of the MH2 domain. The guide RNAs are shown in Key Resources Table. The guide RNAs were expressed from the plasmid pSpCas9(BB)−2A-GFP (pX458) (Addgene, #48138) (Ran et al., 2013). HEK293T and HaCaT were transfected with the appropriate plasmid, and for the double knockout SMAD2 and SMAD3, the two plasmids were transfected simultaneously. Forty-eight hours after transfection, cells were sorted for EGFP expression, plated as single cells in 96-well plates, and screened by Western blot to assess the loss of the protein. For HEK293Ts, two knockout clones for SMAD2, SMAD3, SMAD2/SMAD3, and SMAD4 were used in these studies. For HaCaTs, four independent SMAD4 knockout clones were used. The sequences of the knockout alleles are shown in *Figure 1—source data 1*.

## Knockin of P35S SKI at the endogenous locus

To introduce the P35S mutation into SKI, a gRNA was selected immediately downstream of codon 35. A 120 bp ssODN, where the codon CCG (P35) was mutated to TCC (S35) and codons 33 and 34 where silently mutated from GGC to GGA, was made and purified by Sigma (Key Resources Table). The ssODN contained phosphorothioate bonds between the first two and last two nucleotides at the 5′ and 3′ ends, respectively, to avoid ssODN degradation by endogenous nucleases. The silent mutations at codons 33 and 34 were introduced to increase the specificity of the downstream screening primer. The mutation at codon 35 also disrupts the PAM sequence.

Cells were cotransfected with the pX458 plasmid expressing the gRNA and 10 µM ssODN using Fugene 6. After 48 hr, cells were FACS sorted for GFP expression and plated as single cells in 96-well plates. Subsequently, clones were consolidated, and from replicate plates, genomic DNA was extracted using Quickextract DNA extraction solution (Lucigen) according to manufacturer's instructions. PCR was performed using a universal reverse primer and two different forward primers: a primer which allows detection of WT SKI and one that detects the P35S SKI (Key Resources Table). Clones positive for P35S SKI knockin were selected, verified by Sanger Sequencing, and used for further analysis.

## Western blotting, immunoprecipitations, DNA pulldowns, and immunofluorescence

Whole-cell extracts were prepared as previously described (Inman et al., 2002), while nuclear lysates were prepared according to Wong et al., 1999. Western blots were carried out using standard methods. The list of the antibodies used is shown in Key Resources Table. Immunoprecipitations using GFP-Trap beads (Cromotek) were performed according to the manufacturer's instructions. Immunoprecipitations using antibodies coupled to protein G-Sepharose beads (Sigma) were as described previously (Levy et al., 2007).

DNA pulldown assays were performed as previously described with some modifications (Levy et al., 2007). Nuclear lysates were extracted using buffer containing 360 mM NaCl, and the DNA pulldowns were performed in the presence of a 40 µg of non-biotinylated mutant oligonucleotide to reduce non-specific binding. The oligonucleotides corresponding to WT and mutated SBE of the *JUN* promoter are shown in Key Resources Table. Immunofluorescence was performed as previously described (Pierreux et al., 2000), except that cells were washed and fixed for 5 min in methanol at −20 C (Levy et al., 2007). Nuclei were counter stained with DAPI (0.1 µ/ml). Imaging was performed on a Zeiss Upright 780 confocal microscope. Z-stacks were acquired for all channels, and maximum intensity projection images are shown.

For all of these techniques, a representative experiment of at least two biological repeats is shown.

## Flow cytometry

After 1 hr of TGF-β induction, HaCaT SMAD4 KO rescue cells, expressing either EGFP or EGFP-SMAD4 fusions, were washed, trypsinized, and pelleted. Cell pellets were fixed with methanol for 5 min at −20 C. Fixed cells were incubated with primary antibody against SKIL. Cells were then washed three times in phosphate-buffered saline (PBS) and incubated with secondary antibody conjugated with anti-rabbit Alexa 647. As a negative control, we used cells incubated with secondary antibody only. Antibodies used are listed in Key Resources Table. Subsequently, cells were washed three times with PBS, and pellets were resuspended with 500 μl of PBS and filtered to achieve a single-cell suspension. Cells were then analyzed for EGFP fluorescence on an LSRII flow cytometer (BD Biosciences), gated for viable, single cells. We then quantified the fluorescence emitted by Alexa 647 in cells expressing EGFP as a measure of SKIL protein. The FlowJo program was used to analyze the results.

## Peptide pulldown assays and peptide array

For peptide pulldowns, N-terminal biotinylated peptides were synthesized by the Peptide Chemistry Facility at the Francis Crick Institute using standard procedures. The peptide pulldown assays were performed as described previously (*Randall et al., 2002*). Where recombinant protein was used, it was dissolved in buffer Y (50 mM Tris–HCl, pH 7.5, 150 mM NaCl, 1 mM EDTA, 1% [vol/vol] NP-40) and used at the concentrations given in the legend to *Figure 4*. Peptides sequences are given in Key Resources Table.

A peptide array was generated using peptides corresponding to SKI amino acids 11–45, of which the amino acids 19–35 were mutated one by one to every other amino acid. Arrays were synthesized on an Intavis ResPepSL Automated Peptide Synthesiser (Intavis Bioanalytical Instruments, Germany) on a cellulose membrane by cycles of N(a)-Fmoc amino acids coupling via activation of carboxylic acid groups with diisopropylcarbodiimide in the presence of hydroxybenzotriazole (HOBt), followed by removal of the temporary α-amino protecting group by piperidine treatment. Subsequent to chain assembly, side chain protection groups were removed by treatment of membranes with a deprotection cocktail (20 ml 95% trifluoroacetic acid, 3% triisopropylsilane, 2% water for 4 hr at room temperature) and then washed (4 x dichloromethane, 4 x ethanol, 2 x water, 1 x ethanol), prior to being air dried.

Peptide array membranes were blocked with 5% Milk in 0.01% Tween 20 in PBS and then incubated with purified PSMAD3–SMAD4 complex (see below) overnight. Subsequently, the membranes were washed and incubated with an antibody against SMAD2/3 (BD Biosciences) conjugated to Alexa 488 using a Zenon Mouse IgG Labeling Kit (Life Technology) according to the manufacturer's instructions. Fluorescence was detected where binding occurred between SKI and PSMAD3–SMAD4 complex and measured with a Typhoon FLA 9500 biomolecular imager (GE Healthcare).

In all peptide experiments, a representative of at least two biological repeats is shown.

## Generation of phosphorylated SMAD2 MH2 domain in insect cells

The cDNA encoding a fusion protein consisting of GST followed by a 3C cleavage site and the human SMAD2 MH2 domain (residues 241–465) was inserted into the MCSI of the pFastBac Dual vector (Thermo Fisher Scientific) in the Sal1 and Spe1 restriction sites. The cDNA encoding a constitutively active version of the human TGFBR1 kinase domain (residues 175–503 with a T204D mutation) was cloned into the MCSII using the SmaI and SphI restriction sites. A high-titre baculovirus (>$10^8$ pfu/ml) was generated using standard published protocols (*Fitzgerald et al., 2006*). Expression of the trimeric phosphorylated SMAD2 MH2 domain was performed by infecting Sf21 cells at a density of $1.5 \times 10^6$ cells/ml at a MOI: 1 and incubating for 72 hr at 27°C with rotation at 110 r.p.m. Cells were harvested by centrifugation at $1000 \times$ g for 10 min and stored at −80°C until required.

## Expression of the human phosphorylated SMAD3–SMAD4 complex

An expression construct was generated encoding GST-fused full-length human SMAD3 (with 3C protease site) and inserted into the MCSI of the pFastBac dual. As with the SMAD2 MH2 domain construct, the constitutively active human TGFBR1 kinase domain was cloned into MCSII. Another vector encoding full-length human SMAD4 alone was also constructed, where SMAD4 was inserted

into the pBacPAK plasmid. The resulting vectors were used to generate high-titre virus (>$10^8$ pfu/ml) using a standard published protocol. For expression of the phosphorylated SMAD3–SMAD4 complex both viruses were used to infect cultures of Sf21 insect cells at a density of $1.5 \times 10^6$ cells/ml at a MOI: 1. Infected cultures were allowed to grow for 72 hr at 27°C with rotation at 110 r.p.m. Cells were harvested by centrifugation at 1000 x g for 10 min and stored at −80°C until required.

## Purification of trimeric phosphorylated SMAD2 MH2 domain and phosphorylated SMAD3–SMAD4 complexes

The same procedure was used to purify both phosphorylated SMAD2 MH2 domain trimers and phosphorylated SMAD3–SMAD4 complexes. Typically, 500 ml of infected Sf21 cells were lysed in 30 ml of a lysis buffer consisting of 50 mM HEPES (pH 8.0), 250 mM NaCl, 10% (vol/vol) glycerol, 1% (vol/vol) Triton X-100, 10 mM β-glycerophosphate, 1 mM NaF, 10 mM benzamidine, and 1 mM dithiothreitol (DTT) supplemented with 5 µl BaseMuncher (2500 U/µl), 5 mM MgCl$_2$, and phosphatase (phosphatase inhibitor cocktail 3, Sigma) and protease inhibitors (EDTA-free cOmplete protease inhibitors, Roche). After incubation for 20 min, the suspension was sonicated to ensure complete lysis. The insoluble fraction was pelleted by centrifugation (100,000 × g at 4°C for 30 min). The soluble fraction was incubated with 500 µl bed volume of Glutathione 4B Sepharose (Cytiva) for 2 hr at 4°C with gentle agitation. The resin was washed extensively with buffer containing 50 mM HEPES (pH 7.5), 200 mM NaCl, and 1 mM DTT. The phosphorylated SMAD2 MH2 domain or phosphorylated SMAD3–SMAD4 complexes were eluted from the GSH resin by cleavage with GST-3C protease (20 µg) in 5 ml wash buffer overnight at 4°C with gentle agitation. The proteins were concentrated to 0.5 ml and applied to a S200 10/300 Increase (Cytiva) size exclusion column equilibrated with 50 mM HEPES (pH 7.5), 200 mM NaCl, 5% (vol/vol) glycerol, and 1 mM DTT. Fractions were analyzed by SDS–PAGE and concentrated to 2 mg/ml and snap frozen as 50 µl aliquots and stored at −80°C until required. Purified phosphorylated SMAD2 MH2 domain was quantified using a molar extinction coefficient value of 39,420 $M^{-1}cm^{-1}$. The molar extinction coefficients used for SMAD3 and SMAD4 were 68,870 and 70,820 $M^{-1}cm^{-1}$, respectively.

## Size exclusion chromatography with multiangle laser light scattering

The trimeric arrangement of the phosphorylated SMAD2 MH2 domain was confirmed by SEC-MALLS. In brief, an S200 10/300 Increase column was attached to an AKTA Micro FPLC system (GE Healthcare), which was connected to a Heleos Dawn 8+ followed by an Optilab TRex (Wyatt). For data collection, 100 µl of a 2 mg/ml stock of phosphorylated SMAD2 MH2 domain was injected and data collected at 0.5 ml/min for 60 min. The data were analyzed by ASTRA6.1 software.

## Biolayer interferometry

Biolayer interferometry was carried out using an Octet RED96 instrument (ForteBio). Biotinylated N-SKI peptide (residues 11–45) was immobilized on streptavidin-coated biosensors (ForteBio) at a concentration of 1 µg/ml in buffer containing 50 mM HEPES pH 7.5, 200 mM NaCl, 1 mg/ml bovine serum albumin, and 0.1% Tween-20 for 100 s. The immobilization typically reached a response level of 2 nm. Association and dissociation curves were obtained through addition of a dilution series of trimeric phosphorylated SMAD2 MH2 domain complex (15.6–1.95 µM) for 100 s followed by dissociation in buffer for 350 s using the Octet acquisition software. The binding data were fitted using the Octet analysis software.

## Crystallization of the N-SKI–SMAD2 MH2 domain complex

The SKI peptide (amino acids 11–45) was synthesized by the Peptide Chemistry Group and added in a 2:1 ratio to the phosphorylated SMAD2 MH2 domain trimer. The complex was concentrated to 6 mg/ml and subject to crystallization trials. Initial screening gave rise to fine needles, in several conditions, and these were used as seedstock for rescreening. This gave rise to 25 µm crystals, with a cubic morphology, in the Crystal Screen Cryo screen (Hampton Research); the condition being 1.5 M ammonium sulphate, 0.15 M K Na tartrate, 0.08 M Na$_3$citrate, and 25% (vol/vol) glycerol. The crystals were flash-frozen in liquid nitrogen, and data were collected on the I24 beamline at Diamond Light Source (DLS). Data was processed automatically using the DLS Xia2/XDS pipeline

(*Winter, 2010*; *Kabsch, 2010*). The crystals belong to the $I2_13$ space group and diffracted to a resolution of 2.0 Å.

## Structure determination

Molecular replacement was undertaken with the CCP4 program Phaser (*McCoy et al., 2007*), utilising pdbfile 1khx (with the C-terminal tail removed), as the search model (*Winn et al., 2011*). Initial structure refinement was undertaken with Refmac (*Murshudov et al., 2011*), with manual model building in Coot (*Emsley et al., 2010*), before switching to Phenix.Refine to finalize the model (*Liebschner et al., 2019*; *Afonine et al., 2012*). Coordinates and data are available from the Protein Data Bank, with accession code: 6ZVQ.

## RNA extraction, qRT-PCR, and RNA-sequencing

Total RNA was extracted using Trizol (Thermo Fisher Scientific) according to the manufacturer's instructions. cDNA synthesis and qPCRs were performed as described (*Grönroos et al., 2012*). Primer sequences are listed in Key Resources Table. All qPCRs were performed with the PowerUp SYBR Green Master Mix (Thermo Fisher Scientific) with 300 nM of each primer and 2 μl of diluted cDNA. Fluorescence acquisition was performed on either a 7500 FAST machine or QuantStudio 12 Flex (Thermo Fisher Scientific). Calculations were performed using the ΔΔCt method, and levels of mRNA are expressed as fold change relative to untreated or SB-431542-treated control cells. Means ± SEM from at least three independent experiments are shown. Results were analyzed using Graph-Pad Prism 8 software and statistics were performed on these data using one-way or two-way analysis of variance (ANOVA) as stated in the figure legends.

For the RNA-sequencing experiment four biological replicates were used. Total RNA was extracted, and the quality of the RNA was assessed using a bioanalyzer (Agilent). Libraries were prepared using the KAPA mRNA HyperPrep kit (Roche), and single-end reads were generated using an Illumina HiSeq 4000.

## RNA sequencing analysis

Raw reads were quality and adapter trimmed using cutadapt-1.9.1 (*Martin, 2011*) prior to alignment. Reads were then aligned and quantified using RSEM-1.3.0/STAR-2.5.2 (*Dobin et al., 2013*; *Li and Dewey, 2011*) against the human genome GRCh38 and annotation release 89, both from Ensembl. TPM (transcripts per kilobase million) values were also generated using RSEM/STAR. Differential gene expression analysis was performed in R-3.6.1 (*R Development Core Team, 2009*) using the DESeq2 package (version 1.24.0) (*Love et al., 2014*). Differential genes were selected using a 0.05 false discovery rate threshold using the pairwise comparisons between time points within each cell line individually. Normalization and variance-stabilizing transformation was applied on raw counts before performing principal component analysis and Euclidean distance-based clustering.

For the heatmaps, we selected those genes that were significantly differentially expressed in the control cell line with an absolute log2 (fold change) of at least 0.5, not significantly differentially expressed in the SKI-mutated cell lines, and detected with at least a TPM value of 2 in the control cell lines in any condition. We then visualized the log2 (fold change) for each time point against the control condition. Heatmaps were generated in R-3.6.1 using the ComplexHeatmap (*Gu et al., 2016*) package (version 2.0.0). The raw data files have been uploaded to the European Genome-phenome Archive (EGA), accession number EGAS00001004908.

## Quantifications

Quantification of Western blots was performed by densitometry measurements of each lane using ImageJ software. The measurements were normalized to the loading control in the same blot. In each case, quantifications were normalized to the SB-431542-treated samples. Quantification for the flow cytometry was performed by measurement of the fluorescence emitted by Alexa 647 in EGFP-expressing single cells as a measure of SKIL protein, and levels were normalized to the SB-431542-treated sample in the SMAD4 knockout cells. The program FlowJo was used to analyze the results. For the peptide arrays, two independent experiments were quantified. The staining intensity of each array was quantified by systematically moving a circular selection (diameter 20 pixels) across the

array and measuring the 8-bit greyscale intensity for each spot. Each intensity measure was normalized to the average intensity of 60 positive controls of the WT peptide after subtracting the background, measured from the average intensity of 60 negative controls (truncated SKI peptide C as indicated in *Figure 4—figure supplement 1B*).

### Statistical analysis

Statistical analysis was performed in Prism 8 (GraphPad). At least three independent experiments were performed for statistical analysis unless otherwise specified in the figure legends. Normalized values were log transformed for the statistical analysis. For comparison between more than two groups with one variable, one-way ANOVA was used followed by the Sidak's correction test. For comparison between groups that have been split on two independent variables, two-way ANOVA was performed followed by Tukey's multiple comparison tests.

## Acknowledgements

We would like to thank David Abraham for the normal dermal fibroblasts. We are very grateful to the Francis Crick Institute Advanced Sequencing Facility, the Light Microscopy Facility, the Flow Cytometry Facility, Cell Services, the Genomics Equipment Park and to the Peptide Chemistry Facility. We thank Andrew Economou for quantification of the peptide arrays. We thank all the members of the Hill lab and Davide Coda, Daniel Miller, and Anassuya Ramachandran for helpful discussions and very useful comments on the manuscript. This work was supported by the Francis Crick Institute which receives its core funding from Cancer Research UK (FC001095), the UK Medical Research Council (FC001095), and the Wellcome Trust (FC001095).

## Additional information

### Funding

| Funder | Grant reference number | Author |
| --- | --- | --- |
| Francis Crick Institute | FC10095 | Ilaria Gori<br>Roger George<br>Andrew G Purkiss<br>Stephanie Strohbuecker<br>Rebecca A Randall<br>Roksana Ogrodowicz<br>Dhira Joshi<br>Svend Kjaer<br>Caroline S Hill |

The funders had no role in study design, data collection and interpretation, or the decision to submit the work for publication.

### Author contributions

Ilaria Gori, Conceptualization, Data curation, Formal analysis, Validation, Investigation, Visualization, Writing - original draft, Writing - review and editing; Roger George, Formal analysis, Investigation, Methodology; Andrew G Purkiss, Stephanie Strohbuecker, Data curation, Formal analysis; Rebecca A Randall, Roksana Ogrodowicz, Formal analysis, Investigation; Virginie Carmignac, Laurence Faivre, Resources; Dhira Joshi, Resources, Methodology; Svend Kjær, Supervision, Project administration; Caroline S Hill, Conceptualization, Supervision, Funding acquisition, Writing - original draft, Project administration, Writing - review and editing

### Author ORCIDs

Ilaria Gori https://orcid.org/0000-0003-1913-8520
Roger George https://orcid.org/0000-0002-5547-337X
Andrew G Purkiss https://orcid.org/0000-0002-5931-3509
Stephanie Strohbuecker https://orcid.org/0000-0002-9781-6879
Virginie Carmignac https://orcid.org/0000-0001-8802-6448

Dhira Joshi ID http://orcid.org/0000-0001-8660-2528
Svend Kjær ID https://orcid.org/0000-0001-9767-8683
Caroline S Hill ID https://orcid.org/0000-0002-8632-0480

## Ethics

Human subjects: Dermal fibroblasts from healthy subjects were kindly provided by David Abraham (UCL-Medical School Royal Free Campus) under the ethics of the Health Research Authority, NRES Committee London - Hampstead, Research Ethics Committee (REC) reference, 6398. L32V and ΛS94-97 SKI dermal fibroblasts were obtained from Laurence Faivre and Virginie Carmignac (Université de Bourgogne UMR1231 GAD, Dijon, France) under the ethics of the GAD collection, number DC2011-1332 (Carmignac et al., 2012).

## Decision letter and Author response

Decision letter https://doi.org/10.7554/eLife.63545.sa1
Author response https://doi.org/10.7554/eLife.63545.sa2

## Additional files

### Supplementary files

• Transparent reporting form

### Data availability

Sequencing data have been uploaded to the European Genome-phenome Archive (EGA), accession number EGAS00001004908. Diffraction data have been deposited in PDB under the accession code 6ZVQ. All data generated or analysed during this study are included in the manuscript and supporting files. Source data files have been provided for Figures 1, 2, 4, 5, 6, 7, Figure 1 Supplement 1, Figure 2 Supplement 1, Figure 7 Supplement 2.

The following datasets were generated:

| Author(s) | Year | Dataset title | Dataset URL | Database and Identifier |
|---|---|---|---|---|
| Gori I, George R, Purkiss AG, Strohbuecker S, Randall RA, Ogrodowicz R, Carmignac V, Faivre L, Joshi D, Kjær S, Hill CS | 2020 | Mutations in SKI in Shprintzen-Goldberg syndrome lead to attenuated TGF-$\beta$ responses through SKI stabilization | https://www.ebi.ac.uk/ega/studies/EGAS00001004908 | European Genome-Phenome Archive, EGAS00001004908 |
| Purkiss AG, Kjær S, George R, Hill CS | 2020 | Complex betweenSMAD2 MH2 domain and peptide from Ski corepressor | https://www.rcsb.org/structure/6ZVQ | RCSB Protein Data Bank, 6ZVQ |

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

# Appendix 1

**Appendix 1—key resources table**

| Reagent type (species) or resource | Designation | Source or reference | Identifiers | Additional information |
|---|---|---|---|---|
| Cell line (*Homo sapiens*) | HaCaT | Francis Crick Institute Cell Services | RRID:CVCL_0038 | Keratinocytes (immortalized adult) |
| Cell line (*Homo sapiens*) | HEK293T | Francis Crick Institute Cell Services | RRID:CVCL_0063 | Embryonic kidney cells (normal) |
| Cell line (*Homo sapiens*) | Dermal fibroblasts (normal, adult) | David Abraham, UCL, UK | | Primary cells, male |
| Cell line (*Homo sapiens*) | Dermal fibroblasts (carrying heterozygous mutation L32V SKI) | *Carmignac et al., 2012* PMID:23103230 | | Primary cells, female |
| Cell line (*Homo sapiens*) | Dermal fibroblasts (carrying heterozygous mutation ΔS94-97 SKI) | *Carmignac et al., 2012* PMID:23103230 | | Primary cells, male |
| Cell line (*Homo sapiens*) | HaCaT S4 KO #1; HaCaT S4 KO #2; HaCaT S4 KO #3; HaCaT S4 KO #4 | This paper | | Keratinocytes in which SMAD4 is deleted by CRISPR/Cas9 technology |
| Cell line (*Homo sapiens*) | HEK293T S2 KO #1; HEK293T S2 KO #2 | This paper | | Embryonic kidney cells in which SMAD2 is deleted by CRISPR/Cas9 technology |
| Cell line (*Homo sapiens*) | HEK293T S3 KO #1; HEK293T S3 KO #2 | This paper | | Embryonic kidney cells in which SMAD3 is deleted by CRISPR/Cas9 technology |
| Cell line (*Homo sapiens*) | HEK293T S2/S3 dKO #1; HEK293T S2/S3 dKO #2 | This paper | | Embryonic kidney cells in which SMAD2 and SMAD3 are deleted simultaneously by CRISPR/Cas9 technology |
| Cell line (*Homo sapiens*) | HEK293T S4 KO #1; HEK293T S4 KO #2 | This paper | | Embryonic kidney cells in which SMAD4 is deleted by CRISPR/Cas9 technology |
| Cell line (*Homo sapiens*) | HEK293T P35S SKI #1; HEK293T P35S SKI #2; HEK293T P35S SKI #3; HEK293T P35S SKI #4 | This paper | | Embryonic kidney cells in which the P35S SKI mutation is introduced by CRISPR/Cas9 technology |

*Continued on next page*

*Appendix 1—key resources table continued*

| Reagent type (species) or resource | Designation | Source or reference | Identifiers | Additional information |
|---|---|---|---|---|
| Cell line (*Homo sapiens*) | HEK293T CAGA$_{12}$-Luciferase/Renilla | This paper | | Embryonic kidney cells in which the CAGA$_{12}$-Luciferase and TK-Renilla reporters are stably expressed |
| Cell line (*Homo sapiens*) | HEK293T BRE-Luciferase/Renilla | This paper | | Embryonic kidney cells in which the BRE-Luciferase and TK-Renilla reporters are stably expressed |
| Cell line (*Homo sapiens*) | HEK293T P35S SKI CAGA$_{12}$ Luciferase/Renilla #2; HEK293T P35S SKI CAGA$_{12}$ Luciferase/Renilla #3 | This paper | | HEK293T P35S SKI clones 2 and 3 in which the CAGA$_{12}$-Luciferase and TK-Renilla reporters are stably expressed |
| Cell line (*Homo sapiens*) | HEK293T P35S SKI BRE-Luciferase/Renilla #2; HEK293T P35S SKI BRE-Luciferase/Renilla #3 | This paper | | HEK293T P35S SKI clones 2 and 3 in which the BRE-Luciferase and TK-Renilla reporters are stably expressed |
| Cell line (*Homo sapiens*) | HaCaT S4 KO EGFP | This paper | | HaCaT SMAD4 KO clone 2 cells stably expressing EGFP |
| Cell line (*Homo sapiens*) | HaCaT SMAD4 KO rescue EGFP-SMAD4 WT | This paper | | HaCaT SMAD4 KO clone 2 cells stably expressing EGFP-SMAD4 WT |
| Cell line (*Homo sapiens*) | HaCaT SMAD4 KO rescue EGFP-SMAD4 D351H | This paper | | HaCaT SMAD4 KO clone 2 cells stably expressing EGFP-SMAD4 D351H |
| Cell line (*Homo sapiens*) | HaCaT SMAD4 KO rescue EGFP-SMAD4 D537Y | This paper | | HaCaT SMAD4 KO clone 2 cells stably expressing EGFP-SMAD4 D537Y |
| Cell line (*Homo sapiens*) | HaCaT SMAD4 KO rescue EGFP-SMAD4 A433E | This paper | | HaCaT SMAD4 KO clone 2 cells stably expressing EGFP-SMAD4 A433E |
| Cell line (*Homo sapiens*) | HaCaT SMAD4 KO rescue EGFP-SMAD4 I435Y | This paper | | HaCaT SMAD4 KO clone 2 cells stably expressing EGFP-SMAD4 I435Y |
| Cell line (*M. musculus*) | MEF SMAD2$^{\Delta ex2}$ | *Piek et al., 2001* PMID:11262418 | | Mouse embryo-derived fibroblasts carrying the homozygous null allele *Smad2$^{ex2}$* |
| Cell line (*Spodoptera frugiperda*) | Sf21 | *Fitzgerald et al., 2006* PMID:17117155 | | Insect cells used to express recombinant proteins |

*Continued on next page*

*Appendix 1—key resources table continued*

| Reagent type (species) or resource | Designation | Source or reference | Identifiers | Additional information |
|---|---|---|---|---|
| Transfected construct | pGFP-C1 | VectorBuilder | Clontech, Cat# 6084–1 | Construct used to make HaCaT SMAD4 KO stably expressing EGFP |
| Transfected construct (human) | pEGFP-SMAD4 WT | *Nicolás et al., 2004* PMID:15280432 | | Construct used to make HaCaT SMAD4 KO stably expressing EGFP SMAD4 WT |
| Transfected construct (human) | pEGFP-SMAD4 D351H | This paper | | Construct used to make HaCaT SMAD4 KO stably expressing EGFP-SMAD4 D351H |
| Transfected construct (human) | pEGFP-SMAD4 D537Y | This paper | | Construct used to make HaCaT SMAD4 KO stably expressing EGFP-SMAD4 D537Y |
| Transfected construct (human) | pEGFP-SMAD4 A433E | This paper | | Construct used to make HaCaT SMAD4 KO stably expressing EGFP-SMAD4 A433E |
| Transfected construct (human) | pEGFP-SMAD4 I435Y | This paper | | Construct used to make HaCaT SMAD4 KO stably expressing EGFP-SMAD4 I435Y |
| Transfected construct | pRL-TK Vector | Promega | Cat# E2241 | Construct used to make HEK 293T cells stably expressing TK Renilla |
| Transfected construct | pGL3-BRE-Luc | *Korchynskyi and ten Dijke, 2002* PMID:11729207 | Addgene Cat# 45126 | Construct used to make HEK 293T cells stably expressing Luciferase under control of the BRE |
| Transfected construct | pGL3 CAGA$_{12}$-Luc | *Dennler et al., 1998* PMID:9606191 | | Construct used to make HEK 293T cells stably expressing Luciferase under control of the CAGA$_{12}$ sequence |
| Transfected construct | pSUPER.retro. puro | OligoEngine | Cat# VEC-pRT-0002 | Construct used to express resistance to puromycin in order to make stable cell lines |
| Transfected construct | pSpCas9(BB)—2A-GFP (PX458) | *Ran et al., 2013* PMID:24157548 | Addgene Cat# 48138 | Different oligonucleotides corresponding to gRNAs have been cloned into this plasmid in order to make several different CRISPR/Cas9 knockout cell lines. |
| Antibody | Anti-phosphorylated SMAD2 (Rabbit monoclonal) | Cell Signaling Technology | Cat# 3108; RRID:AB_490941 | WB (1:1000) |
| Antibody | Anti-SMAD2/3 (mouse monoclonal) | BD Biosciences | Cat# 610843; RRID:AB_398162 | IF (1:500) WB (1:1000) |
| Antibody | Anti-phospho-SMAD3 (rabbit monoclonal) | Cell Signaling Technology | Cat# 9520; RRID:AB_2193207 | WB (1:500) |
| Antibody | Anti-SMAD4 (B-8) (mouse monoclonal) | Santa Cruz | Cat# sc-7966; RRID:AB_627905 | WB (1:1000) |
| Antibody | Anti-SMAD3 (Rabbit monoclonal) | Abcam | Cat# 40854; RRID:AB_777979 | WB (1:1000) |

*Continued on next page*

*Appendix 1—key resources table continued*

| Reagent type (species) or resource | Designation | Source or reference | Identifiers | Additional information |
|---|---|---|---|---|
| Antibody | Anti-FLAG DYKDDDDK Tag (L5) (Rat monoclonal) | Thermo Fisher | Cat# MA1-142, RRID:AB_2536846 | IP (5 μg) WB (1:1000) |
| Antibody | Anti-MCM6 (C-20) (Goat polyclonal) | Santa Cruz | Cat# sc-9843; RRID:AB_2142543 | WB (1:2000) |
| Antibody | Anti MCM6 (H-8) (Mouse monoclonal) | Santa Cruz | Cat# sc-393618 RRID:AB_2885187 | WB (1:2000) |
| Antibody | Anti-Tubulin (Rat monoclonal) | Abcam | Cat# ab6160; RRID:AB_305328 | WB (1:5000) |
| Antibody | Anti-SKI (Rabbit Polyclonal) | GeneTex | Cat# GTX133764 RRID:AB_2885186 | IP (5 μg) WB (1:2000) |
| Antibody | Anti-GFP (Goat polyclonal) | Abcam | Cat# ab6673, RRID:AB_305643 | IF (1:200) |
| Antibody | Anti SKIL (SnoN) (H-317) (Rabbit polyclonal) | Santa Cruz | Cat# sc-9141, RRID:AB_671124 | WB (1:1000) IF (1:1000) Flow cytometry (1:200) |
| Antibody | Donkey anti-Goat IgG (H+L) Cross-adsorbed secondary antibody, Alexa Fluor 546 | Thermo Fisher Scientific | Cat# A-11056, RRID:AB_2534103 | IF (1:1000) |
| antibody | Anti-Rabbit Alexa Fluor 594 | Thermo Fisher Scientific | Cat# A-21244; RRID:AB_10562581 | IF (1:1000) |
| Antibody | Anti-Rabbit Alexa Fluor 647 | Thermo Fisher Scientific | Cat# A-21244, RRID:AB_2535812 | Flow cytometry (1:1000) |
| Antibody | Goat anti-rabbit HRP | Dako | Cat# P0448; RRID:AB_2617138 | WB (1:5000) |
| Antibody | Goat anti-mouse HRP | Dako | Cat#P0447; RRID:AB_2617137 | WB (1:5000) |
| Antibody | Donkey anti-rat HRP | Jackson Lab | Cat#712-035-153; RRID:AB_2340639 | WB (1:5000) |
| Antibody | Rabbit anti-goat HRP | Dako | Cat# P0449, RRID:AB_2617143 | WB (1:5000) |
| Recombinant DNA reagent | pEF-FLAG-SKIL (plasmid) | This paper | | Transient transfection in HEK293T cells to express FLAG SKIL WT |
| Recombinant DNA reagent | FLAG SKIL G103V (SKIL ΔS2/S3) (plasmid) | This paper | | Transient transfection in HEK293T cells to express FLAG SKIL G103V |

*Continued on next page*

*Appendix 1—key resources table continued*

| Reagent type (species) or resource | Designation | Source or reference | Identifiers | Additional information |
|---|---|---|---|---|
| Recombinant DNA reagent | FLAG SKIL R314A, T315A, H317A, W318E (SKIL ΔS4) (plasmid) | This paper | | Transient transfection in HEK293T cells to express FLAG SKIL R314A, T315A, H317A, W318E |
| Recombinant DNA reagent | pFast Dual ALK5* GST-MH2-hSMAD2 (plasmid) | This paper | | Used to express recombinant proteins in insect cells |
| Recombinant DNA reagent | pFastDual ALK5*/GST-hSMAD3 (plasmid) | This paper | | Used to express recombinant proteins in insect cells |
| Recombinant DNA reagent | pBacPAK-SMAD4 (plasmid) | This paper | | Used to express recombinant proteins in insect cells |
| Sequence-based reagent | SMAD4_F1 | This paper | | CACCGACAACTCGTTCGTAGTGATA CRISPR/Cas9-mediated knockout guide, forward oligo 1 |
| Sequence-based reagent | SMAD4_R1 | This paper | | AAACTATCACTACGAACGAGTTGTC CRISPR/Cas9 mediated knockout guide, reverse oligo 1 |
| Sequence-based reagent | SMAD4_F2 | This paper | | CACCGTGAGTATGCATAAGCGACGA CRISPR/Cas9 mediated knockout guide, forward oligo 2 |
| Sequence-based reagent | SMAD4_R2 | This paper | | AAACTCGTCGCTTATGCATACTCAC CRISPR/Cas9 mediated knockout guide, reverse oligo 2 |
| Sequence-based reagent | SMAD2 _F1 | This paper | | CACCGCTATCGAACACCAAAATGC CRISPR/Cas9 mediated knockout guide, forward oligo |
| Sequence-based reagent | SMAD2 _R1 | This paper | | AAACGCATTTTGGTGTTCGATAGC CRISPR/Cas9 mediated knockout guide, reverse oligo |
| Sequence-based reagent | SMAD3_F1 | This paper | | CACCGGAATGTCTCCCCGACGCGC CRISPR/Cas9 mediated knockout guide, forward oligo |
| Sequence-based reagent | SMAD3_R1 | This paper | | AAACGCGCGTCGGGGAGACATTCC CRISPR/Cas9 mediated knockout guide, reverse oligo |
| Sequence-based reagent | SKI_F1 | This paper | | CACCGCAGCGCGCCGAGAAAGCGGC CRISPR/Cas9 mediated knockin guide, forward oligo |
| Sequence-based reagent | SKI_R1 | This paper | | AAACGCCGCTTTCTCGGCGCGCTGC CRISPR/Cas9 mediated knockin guide, reverse oligo |
| Sequence-based reagent | SKI_ssODN | This paper | | G*C*AGTTCCACCTGAGCTCCATGAGC TCGCTGGGAGGATCCGCCGCTTTCTC GGCGCGCTGGGCGCAGGAGGCCTA CAAGAAGGAGAGCGCCAAGGAGGCG GGCGCGGCCGCGGTGCCG*G*C Repair template |

*Continued on next page*

*Appendix 1—key resources table continued*

| Reagent type (species) or resource | Designation | Source or reference | Identifiers | Additional information |
|---|---|---|---|---|
| Sequence-based reagent | SKI_R2 | This paper | | GCCCATGACTTTGAGGATCTCC Universal reverse primer for clone screening |
| Sequence-based reagent | SKI_F2 | This paper | | ATGAGCTCGCTGGGCGGCCCG WT SKI forward primer for clone screening |
| Sequence-based reagent | SKI_F3 | This paper | | ATGAGCTCGCTGGGAGGATCC P35S SKI forward primer for clone screening |
| Sequence-based reagent | Biotinylated WT cJUN SBE oligonucleotide (top) | *Levy et al., 2007* PMID:17591695 | | 5′-Biotin-GGAGGTGCGCGGAGTCAGG CAGACAGACAGACACAGCCA GCCAGCCAGGTCGGCA DNAP oligo - Top |
| Sequence-based reagent | WT cJUN SBE oligonucleotide (bottom) | *Levy et al., 2007* PMID:17591695 | | TGCCGACCTGGCTGGCTGGC TGTGTCTGTCTGTCTGCCTG ACTCCGCGCACCTCC DNAP oligo - Bottom |
| Sequence-based reagent | Biotinylated MUT cJUN SBE oligonucleotide (top) | This paper | | 5′-biotin-GGATTTGCTAATGATATAGT AATATATATATATACATATAT ATATATTGATCTTCA Mutated DNAP oligo -Top |
| Sequence-based reagent | MUT cJUN SBE oligonucleotide (bottom) | This paper | | TGAAGATCAATATATATATAT GTATATATATATATTACTAT ATCATTAGCAAATCC Mutated DNAP oligo -Bottom |
| Sequence-based reagent | MUT cJUN SBE oligo-nucleotide (top) | This paper | | GGATTTGCTAATGATATAGTA ATATATATATATACATATAT ATATATTGATCTTCA Competitor mutant oligo - Top |
| Sequence-based reagent | MUT cJUN SBE oligo-nucleotide (bottom) | This paper | | TGAAGATCAATATATATATATG TATATATATATTACTATA TCATTAGCAAATCC Competitor mutant oligo – bottom |
| Sequence-based reagent | GAPDH_F1 | *Grönroos et al., 2012* PMID:22615489 | | CTTCAACAGCGACACCCACT PCR primer |
| Sequence-based reagent | GAPDH_R1 | *Grönroos et al., 2012* PMID:22615489 | | GTGGTCCAGGGGTCTTACTC PCR primer |
| Sequence-based reagent | SMAD7_F1 | *Grönroos et al., 2012* PMID:22615489 | | CTTAGCCGACTCTGCGAACT PCR primer |
| Sequence-based reagent | SMAD7_R1 | *Grönroos et al., 2012* PMID:22615489 | | CCAGGCTCCAGAAGAAGTTG PCR primer |
| Sequence-based reagent | SKIL_F1 | This paper | | CTGGGGCTTTGAATCAGCTA PCR primer |
| Sequence-based reagent | SKIL_R1 | This paper | | CATGGTCACCTTCCTGCTTT PCR primer |

*Continued on next page*

*Appendix 1—key resources table continued*

| Reagent type (species) or resource | Designation | Source or reference | Identifiers | Additional information |
|---|---|---|---|---|
| Sequence-based reagent | SERPINE1_F1 | *Grönroos et al., 2012* PMID:22615489 | | TGATGGCTCAGACCAACAAG PCR primer |
| Sequence-based reagent | SERPINE1_R1 | *Grönroos et al., 2012* PMID:22615489 | | GTTGGTGAGGGCAGAGAGAG PCR primer |
| Sequence-based reagent | JUNB_F1 | *Ramachandran et al., 2018* PMID:29376829 | | ATACACAGCTACGGGATACGG PCR primer |
| Sequence-based reagent | JUNB_R1 | *Ramachandran et al., 2018* PMID:29376829 | | GCTCGGTTTCAGGAGTTTGT PCR primer |
| Sequence-based reagent | ID1_F1 | *Ramachandran et al., 2018* PMID:29376829 | | GCCGAGGCGGCATGCGTTC PCR primer |
| Sequence-based reagent | ID1_R1 | *Ramachandran et al., 2018* PMID:29376829 | | CTTGCCCCCTGGATGGCTGG PCR primer |
| Sequence-based reagent | ID3_F1 | *Grönroos et al., 2012* PMID:22615489 | | GGCCCCCACCTTCCCATCC PCR primer |
| Sequence-based reagent | ID3_R1 | *Grönroos et al., 2012* PMID:22615489 | | GCCAGCACCTGCGTTCTGGAG PCR primer |
| Sequence-based reagent | CDKN1A_F1 | *Miller et al., 2018* PMID:30428352 | | ACTCTCAGGGTCGAAAACGG PCR primer |
| Sequence-based reagent | CDKN1A_R1 | *Miller et al., 2018* PMID:30428352 | | ATGTAGAGCGGGCCTTTGAG PCR primer |
| Sequence-based reagent | ISLR2_F1 | This paper | | AGTCGGCGAATATTGGGAGC PCR primer |
| Sequence-based reagent | ISLR2_R1 | This paper | | ATGATCCGGCCACTCCTAGA PCR primer |
| Sequence-based reagent | CALB2_F1 | This paper | | ATGGCAAATTGGGCCTCTCA PCR primer |
| Sequence-based reagent | CALB2_R1 | This paper | | GGTCAGCTTCATGCCCTGAAAT PCR primer |
| Sequence-based reagent | SOX11_F1 | This paper | | AGCGGAGGAGGTTTTCAGTG PCR primer |
| Sequence-based reagent | SOX11_R1 | This paper | | TTCCATTCGGTCTCGCCAAA PCR primer |
| Sequence-based reagent | ITGB6_F1 | This paper | | TGCGACCATCAGTGAAGAAG PCR primer |
| Sequence-based reagent | ITGB6_R1 | This paper | | GACAACCCCGATGAGAAGAA PCR primer |

*Appendix 1—key resources table continued*

| Reagent type (species) or resource | Designation | Source or reference | Identifiers | Additional information |
|---|---|---|---|---|
| Sequence-based reagent | HEY1_F1 | This paper | | GCTTTTGAGAAGCAGGGATCT PCR primer |
| Sequence-based reagent | HEY1_R1 | This paper | | GATAACGCGCAACTTCTGCC PCR primer |
| Sequence-based reagent | COL7A1_F1 | This paper | | CAAGGGGGACATGGGTGAAC PCR primer |
| Sequence-based reagent | COL7A1_R1 | This paper | | CGGATACCAGGCACTCCATC PCR primer |
| Sequence-based reagent | SMAD4_F3 | This paper | | GTTCATAAGATCTACCCAAGTG AATATATAAAGGTCTTTGATTTG PCR primer for cloning, forward primer carrying the mutation A433E |
| Sequence-based reagent | SMAD4_R3 | This paper | | ACGCAAATCAAAGACCTTTATA TATTCACTTGGGTAGATCTTATG PCR primer for cloning, reverse primer carrying the mutation A433E |
| Sequence-based reagent | SMAD4_F4 | This paper | | AAGATCTACCCAAGTGCATATT ACAAGGTCTTTGATTTGCGTCAG PCR primer for cloning, forward primer carrying the mutation I435Y |
| Sequence-based reagent | SMAD4_R4 | This paper | | CTGACGCAAATCAAAGACCTTGT AATATGCACTTGGGTAGATCTT PCR primer for cloning, reverse primer carrying the mutation I435Y |
| Sequence-based reagent | SKIL_F2 | This paper | | CAGAGCTCGCTGGGTGTA CCAGCAGCATTTTC PCR primer for cloning, forward primer carrying the mutation G103V |
| Sequence-based reagent | SKIL_R2 | This paper | | GAAAATGCTGCTGGTAC ACCCAGCGAGCTCTG PCR primer for cloning, reverse primer carrying the mutation G103V |
| Peptide, recombinant protein | SKI peptide A | This paper | | Biotin-aminohexanoic acid-FQPHPGLQKTLEQFHLSSMSS LGGPAAFSARWAQEAYKKES |
| Peptide, recombinant protein | SKI peptide B | This paper | | Biotin-aminohexanoic acid-FQPHPGLQKTLEQFHLS SMSSLGGPAAFSARWAQE |
| Peptide, recombinant protein | SKI peptide C | This paper | | Biotin-aminohexanoic acid-GLQKTLEQFHLSSMSSLG GPAAFSARWAQE |

*Appendix 1—key resources table continued*

| Reagent type (species) or resource | Designation | Source or reference | Identifiers | Additional information |
|---|---|---|---|---|
| Peptide, recombinant protein | SKI peptide D | This paper | | Biotin-aminohexanoic acid-TLEQFHLSSMSSLGG PAAFSARWAQE |
| Peptide, recombinant protein | SKI peptide E | This paper | | Biotin-aminohexanoic acid-TLEQFHLSSMSSLGGPA AFSARWAQEAYK |
| Peptide, recombinant protein | SKI L21R | This paper | | Biotin-aminohexanoic acid-FQPHPGLQKTREQFHLSS MSSLGGPAAFSARWAQE |
| Peptide, recombinant protein | SKI S28T | This paper | | Biotin-aminohexanoic acid-FQPHPGLQKTLEQFHLS TMSSLGGPAAFSARWAQE |
| Peptide, recombinant protein | SKI S31L | This paper | | Biotin-aminohexanoic acid-FQPHPGLQKTLEQFHLS SMSLLGGPAAFSARWAQE |
| Peptide, recombinant protein | SKI L32P | This paper | | Biotin-aminohexanoic acid-FQPHPGLQKTLEQFHLS SMSSPGGPAAFSARWAQE |
| Peptide, recombinant protein | SKI G34D | This paper | | Biotin-aminohexanoic acid – FQPHPGLQKTLEQFHLSS MSSLGDPAAFSARWAQE |
| Peptide, recombinant protein | SKI P35S | This paper | | Biotin-aminohexanoic acid – FQPHPGLQKTLEQFHLSS MSSLGGSAAFSARWAQE |
| Peptide, recombinant protein | SKIL WT | This paper | | Biotin-aminohexanoic acid – LHLNPSLKHTLAQFHLSSQSS LGGPAAFSARHSQESMSPTV |
| Peptide, recombinant protein | SKIL L90R | This paper | | Biotin-aminohexanoic acid – LHLNPSLKHTRAQFHLSSQS SLGGPAAFSARHSQESMSPTV |
| Peptide, recombinant protein | SKIL S100L | This paper | | Biotin-aminohexanoic acid - LHLNPSLKHTLAQFHLSSQS LLGGPAAFSARHSQESMSPTV |
| Peptide, recombinant protein | SKIL G103D | This paper | | Biotin-aminohexanoic acid - LHLNPSLKHTLAQFHLSSQSS LGDPAAFSARHSQESMSPTV |
| Peptide, recombinant protein | SKIL P104S | This paper | | Biotin-aminohexanoic acid - LHLNPSLKHTLAQFHLSSQSS LGGSAAFSARHSQESMSPTV |
| Peptide, recombinant protein | SKI WT | This paper | | FQPHPGLQKTLEQFHLSSMSS LGGPAAFSARWAQE |

*Continued on next page*

*Appendix 1—key resources table continued*

| Reagent type (species) or resource | Designation | Source or reference | Identifiers | Additional information |
|---|---|---|---|---|
| Peptide, recombinant protein | Human recombinant TGF-β1 | Peprotech | Cat# 100–21 | |
| Peptide, recombinant protein | Human recombinant BMP4 | Peprotech | Cat# 120-05ET | |
| Peptide, recombinant protein | Human recombinant Activin A | Peprotech | Cat# 120–14 | |
| Commercial assay or kit | PowerUp SYBR Green Master Mix | Thermo Fisher Scientific | Cat# A25742 | |
| Commercial assay or kit | Quickextract DNA extraction solution | Lucigen | Cat# QE09050 | |
| Commercial assay or kit | Dual-Glo Luciferase Assay System | Promega | Cat# E2920 | |
| Commercial assay or kit | Fugene 6 transfection reagent | Promega | Cat# E2691 | |
| Commercial assay or kit | Superdex 200 10/300 size exclusion column | Cytiva | Cat# 28990944 | |
| Commercial assay or kit | Streptavidin (SA) Biosensors | ForteBio | Cat# 18–5019 | |
| Commercial assay or kit | Glutathione 4B Sepharose | Cytiva | Cat# 17075601 | |
| Commercial assay or kit | Pierce NeutrAvidin Agarose | Thermo Fisher Scientific | Cat# 29200 | |
| Commercial assay or kit | GFP-Trap Agarose (IP) | Cromotek | Cat# gta-20 | |
| Commercial assay or kit | Protein G Sepharose, Fast Flow | Sigma–Aldrich | Cat# P3296 | |
| Chemical compound, drug | cOmplete, EDTA-free Protease Inhibitor Cocktail | Sigma–Aldrich | Cat# 000000011873580001 | |
| Chemical compound, drug | TRIzol | Thermo Fisher Scientific | Cat# 15596026 | |
| Chemical compound, drug | DAPI | Sigma-Aldrich | Cat# 10236276001 | |
| Chemical compound, drug | SB-431542 | Tocris; *Inman et al., 2002*: PMID:12065756 | Cat# 1614 | |

*Continued on next page*

*Appendix 1—key resources table continued*

| Reagent type (species) or resource | Designation | Source or reference | Identifiers | Additional information |
|---|---|---|---|---|
| Chemical compound, drug | MG-132 | Tocris | Cat# 1748 | |
| Software, algorithm | FIJI (ImageJ) | https://imagej.net/Fiji/Downloads | RRID:SCR_002285 | |
| Software, algorithm | FlowJo 10 | FlowJo | RRID:SCR_008520 | |
| Software, algorithm | GraphPad Prism 8 | GraphPad | RRID:SCR_002798 | |
| Software, algorithm | ASTRA6.1 | Wyatt | RRID:SCR_001625 | |
| Software, algorithm | Octet CFR software | ForteBio | | |
| Software, algorithm | DLS Xia2/XDS pipeline | *Winter, 2010*; *Kabsch, 2010* PMID:20124692 | | |
| Software, algorithm | CCP4 suite | *Winn et al., 2011* PMID:21460441 | RRID:SCR_007255 | |
| Software, algorithm | Phaser | *McCoy et al., 2007* PMID:19461840 | RRID:SCR_014219 | |
| Software, algorithm | Refmac | *Murshudov et al., 2011* PMID:21460454 | RRID:SCR_014225 | |
| Software, algorithm | Coot | *Emsley et al., 2010* PMID:20383002 | RRID:SCR_014222 | |
| Software, algorithm | PHENIX Suite | *Liebschner et al., 2019* PMID:31588918 | RRID:SCR_014224 | |
| Software, algorithm | Phenix.Refine | *Afonine et al., 2012* PMID:22505256 | RRID:SCR_016736 | |
| Software, algorithm | RSEM-1.3.0/ STAR-2.5.2 | *Li and Dewey, 2011*; PMID:21816040 *Dobin et al., 2013* PMID:23104886 | RRID:SCR_013027 | |
| Software, algorithm | DESeq2 package (version 1.24.0) | *Love et al., 2014* PMID:25516281 | RRID:SCR_015687 | |

