## [Decision Letter]

**Acceptance summary:**

Shprintzen-Goldberg syndrome (SGS) is a multi-systemic connective tissue disorder, with considerable clinical overlap with Marfan and Loeys-Dietz syndromes. These syndromes have commonly been associated with enhanced TGF-β signaling. However, new data from the Hill laboratory reveal that SGS is associated with an attenuation of TGF-β-induced transcriptional responses (not an enhancement) and provide insight into the molecular mechanism. This work has important implications for the treatment of Marfan-related syndromes.

**Decision letter after peer review:**

Thank you for submitting your article "Mutations in SKI in Shprintzen-Goldberg syndrome lead to attenuated TGF-β responses through SKI stabilization" for consideration by *eLife*. Your article has been reviewed by three peer reviewers, one of whom is a member of our Board of Reviewing Editors, and the evaluation has been overseen by Philip Cole as the Senior Editor.

The reviewers have discussed the reviews with one another and the Reviewing Editor has drafted this decision to help you prepare a revised submission.

Summary:

This is an interesting manuscript that demonstrates that SGS-associated mutations in the transcriptional repressor Ski prevent TGFb-induced degradation of Ski. This Ski stabilization is associated with suppression of binding to SMAD2/3 while maintaining repression through an interaction with SMAD4. The conclusions of this study serve to extend established knowledge of SGS-associated Ski mutations and clarify several conflicting/controversial observations related to Marfan-related syndromes more broadly.

In general, this is a strong study and presents good biochemical data using cells with CRISPR-mediated knock-in alleles to mimic SGS-associated Ski mutations as well as patient-derived cell lines. No major criticisms of the study were noted by the reviewers with the exception of a question concerning the structural analysis (Figure 5).

Essential revisions:

1) The crystal structure presented shows an N-terminal Ski peptide in a complex with SMAD2. It is unclear what new information this structure presents since a structure of Ski in complex with Smad2 has previously been reported (Mizazono et al., 2018). The authors should state what conclusions are drawn based on the new data that were not previously based on the Mizazono et al. analysis. It is not obvious that the new structure presented provides new information – if it does, this should be explicitly stated. The presentation and analysis of the structure in the Results and Discussion section of the manuscript should appropriately refer to Mizazono et al. At present, Mizazono et al. is only obliquely cited in the Discussion section of the manuscript.

---

## [Author Response]

Essential revisions:1) The crystal structure presented shows an N-terminal Ski peptide in a complex with SMAD2. It is unclear what new information this structure presents since a structure of Ski in complex with Smad2 has previously been reported (Mizazono et al., 2018). The authors should state what conclusions are drawn based on the new data that were not previously based on the Mizazono et al. analysis. It is not obvious that the new structure presented provides new information – if it does, this should be explicitly stated. The presentation and analysis of the structure in the Results and Discussion section of the manuscript should appropriately refer to Mizazono et al. At present, Mizazono et al. is only obliquely cited in the Discussion section of the manuscript.

We apologize for not referring to the SMAD2–SKI structure published by Miyazono et al., in 2018, in the Results. We have now remedied this. We had actually produced our structure before the publication by Miyazono et al. We think it is very important to publish our structure in this paper, as to our knowledge it is the first SMAD structure to use eukaryotic-cell produced SMAD protein that has been phosphorylated in vivo by co-expression of the kinase domain of the relevant receptor (TGFBR1 in this case). This contrasts with the Miyazono structure that used *E. coli* produced pseudo-phosphorylated SMAD2 MH2 domain (where glutamic acids are used to mimic phosphates). In addition, our SKI peptide was not tagged, whereas the peptide used for the Miyazono structure has an acidic C-terminal tag. The main features of the structure are similar. This has now been clarified in the Results. We have also referenced the Miyazono paper in the Discussion.

Another advantage of our structure of the phosphorylated SMAD2 MH2 domain is that we have been able to use it to understand why SKI and SKIL only bind to phosphorylated SMAD2 and not to monomeric SMAD2. This issue was not addressed in Miyazono et al., 2018. This too has been clarified in the Results.